# Analyzing Feed-Forward Blocks in Transformers through the Lens of Attention Maps

**Goro Kobayashi**[1,3]**, Tatsuki Kuribayashi**[2,1]**, Sho Yokoi**[1,3]**, Kentaro Inui**[2,1,3]
[1]Tohoku University, [2]MBZUAI, [3]RIKEN
goro.koba@dc.tohoku.ac.jp    tatsuki.kuribayashi@mbzuai.ac.ae
yokoi@tohoku.ac.jp    kentaro.inui@mbzuai.ac.ae

## Abstract

Transformers are ubiquitous in wide tasks. Interpreting their internals is a pivotal goal. Nevertheless, their particular components, feed-forward (FF) blocks, have typically been less analyzed despite their substantial parameter amounts. We analyze the input contextualization effects of FF blocks by rendering them in the attention maps as a human-friendly visualization scheme. Our experiments with both masked- and causal-language models reveal that FF networks modify the input contextualization to emphasize specific types of linguistic compositions. In addition, FF and its surrounding components tend to cancel out each other's effects, suggesting potential redundancy in the processing of the Transformer layer.

⬤ github.com/gorokoba560/norm-analysis-of-transformer

## 1 Introduction

Transformer is composed of several components (e.g., self-attention and feed-forward networks); tracking and interpreting its *component-by-component intermediate processing* have been aimed to enrich the mechanistic interpretation of the model's inner workings (Kobayashi et al. 2020; 2021; Modarressi et al. 2022; see § A.1 for more works). One straightforward yet popular approach is to render vanilla attention weights, reflecting how strongly a particular context token contributes to computing an output representation (§ 2), and identify in which attention head a specific type of contextualization is performed in Transformer (Clark et al., 2019; Kovaleva et al., 2019). Recent studies have extended this approach by incorporating more components beyond Query-Key matrix multiplication, e.g., residual connections, enabling to analyze how input-contextualization is shaped component-by-component through the lens of attention maps (Brunner et al. 2020; Abnar & Zuidema 2020; Kobayashi et al. 2020; 2021; Modarressi et al. 2022; see § A.2 for more works). Such an interpretation scheme has several advantages: (i) Attention mechanisms are spread out in the entire Transformer architecture; thus, if one can view their surrounding component's processing through this "semi-transparent window," these pictures can complete the Transformer's entire *component-by-component* internal processing, (ii) token-to-token relationships are more familiar to humans than directly observing high-dimensional, continuous intermediate representations/parameters, and (iii) input attribution is of major interest in explaining the model prediction, and (iv) such attention map refinement approaches tend to estimate better attributions than, e.g., gradient-based methods (Modarressi et al., 2022; 2023).

Nevertheless, in this attention-map refinement approach, FF networks have typically been overlooked from the analysis scope, although there are several motivations to consider FFs. For example, FFs account for about two-thirds of the layer parameters in typical Transformer-based models, such as BERT and GPT series; this implies that FF has the expressive power to dominate the model's inner workings. In addition, there is a growing general interest in FFs with the rise of FF-focused methods such as adapters (Houlsby et al., 2019), although analyzing these specially designed models is beyond this paper's focus. Furthermore, it has been reported that FFs indeed perform some linguistic operations, while existing studies have not explicitly focused on input contextualization (Geva et al., 2021; Dai et al., 2022). For example, Oba et al. (2021) explored the relationship between FF's neuron activation and specific phrases; Geva et al. (2021), Meng et al. (2022), and Dai et al. (2022) examined the knowledge stored in FF's parameters through viewing FF as key-value memories.

In this study, we analyze the *FF blocks* in the Transformer layer, i.e., FF networks and their surrounding residual and normalization layers, with respect to their impact on *input contextualization* through the lens of attention maps. Notably, although the FFs are applied to each input representation independently, their transformation can inherently affect the input contextualization (§ 3), and our experiments show that this indeed occurs (§ 5.1). Technically, we propose a method to compute attention maps reflecting the FF blocks' processing by extending a norm-based analysis (Kobayashi et al., 2020; 2021), which has several advantages: the impact of input ($\|\boldsymbol{x}\|$) is considered unlike the vanilla gradient, and that only the forward computation is required. Although the original norm-based approach can not be simply applied to the non-linear part in the FF, this study handles this limitation by partially applying an integrated gradient (Sundararajan et al., 2017) and enables us to track the input contextualization from the information geometric perspectives.

Our experiments with both masked- and causal-language models (LMs) disclosed the contextualization effects of the FF blocks. Specifically, we first reveal that FF and layer normalization in specific layers tend to largely control contextualization. We also observe typical FF effect patterns, independent of the LM types, such as amplifying a specific type of lexical composition, e.g., subwords-to-word and words-to-multi-word-expression constructions (§ 5.2). Furthermore, we also first observe that the FF's effects are weakened by surrounding residual and normalization layers (§ 6), suggesting redundancy in the Transformer's internal processing.

## 2 BACKGROUND

**Notation:** Boldface letters such as $\boldsymbol{x}$ denote row vectors.

**Transformer layer:** The Transformer architecture (Vaswani et al., 2017) consists of a series of layers, each of which updates each token representation $\boldsymbol{x}_i \in \mathbb{R}^d$ in input sequence $\boldsymbol{X} := [\boldsymbol{x}_1^\top, \ldots, \boldsymbol{x}_n^\top]^\top \in \mathbb{R}^{n \times d}$ to a new representation $\boldsymbol{y}_i \in \mathbb{R}^d$. That is, the information of context $\boldsymbol{X}$ is added to $\boldsymbol{x}_i$, and $\boldsymbol{x}_i$ is updated to $\boldsymbol{y}_i$. We call this process contextualization of $\boldsymbol{x}_i$. Each layer is composed of four parts: multi-head attention (ATTN), feed-forward network (FF), residual connection (RES), and layer normalization (LN) (see Fig. 1). Note that we use the Post-LN architecture (Vaswani et al., 2017) for the following explanations, but our methods can simply be extended to the Pre-LN variant (Xiong et al., 2020). A single layer can be written as a composite function: $\boldsymbol{y}_i = \text{Layer}(\boldsymbol{x}_i; \boldsymbol{X}) = (\text{FFB} \circ \text{ATB})(\boldsymbol{x}_i; \boldsymbol{X}) = (\text{LN2} \circ \text{RES2} \circ \text{FF} \circ \text{LN1} \circ \text{RES1} \circ \text{ATTN})(\boldsymbol{x}_i; \boldsymbol{X})$. We call $(\text{LN1} \circ \text{RES1} \circ \text{ATTN})(\cdot)$ attention block (ATB), and $(\text{LN2} \circ \text{RES2} \circ \text{FF})(\cdot)$ feed-forward block (FFB). Each component updates the representation as follows:

$$\text{ATTN}(\boldsymbol{x}_i; \boldsymbol{X}) = (\sum_h \sum_j \alpha_{i,j}^h \boldsymbol{x}_j \boldsymbol{W}_V + \boldsymbol{b}_V) \boldsymbol{W}_O + \boldsymbol{b}_O \in \mathbb{R}^d \tag{1}$$

$$\text{where} \quad \alpha_{i,j}^h := (\boldsymbol{x}_i \boldsymbol{W}_Q + \boldsymbol{b}_Q)(\boldsymbol{x}_i \boldsymbol{W}_K + \boldsymbol{b}_K)^\top \in \mathbb{R} \tag{2}$$

$$\text{FF}(\boldsymbol{z}_i) = \boldsymbol{g}(\boldsymbol{z}_i \boldsymbol{W}_1 + \boldsymbol{b}_1) \boldsymbol{W}_2 + \boldsymbol{b}_2 \in \mathbb{R}^d \tag{3}$$

$$(\text{RES} \circ \boldsymbol{f})(\boldsymbol{z}_i) = \boldsymbol{f}(\boldsymbol{z}_i) + \boldsymbol{z}_i \in \mathbb{R}^d \tag{4}$$

$$\text{LN}(\boldsymbol{z}_i) = \frac{\boldsymbol{z}_i - m(\boldsymbol{z}_i)}{s(\boldsymbol{z}_i)} \odot \boldsymbol{\gamma} + \boldsymbol{\beta} \in \mathbb{R}^d, \tag{5}$$

where $\boldsymbol{W}, \boldsymbol{\gamma}$ denote weight parameters, and $\boldsymbol{b}, \boldsymbol{\beta}$ denote bias parameters corresponding to query (Q), key (K), value (V), etc. $\boldsymbol{f} \colon \mathbb{R}^d \to \mathbb{R}^d$, $\boldsymbol{g} \colon \mathbb{R}^{d'} \to \mathbb{R}^{d'}$, $m \colon \mathbb{R}^d \to \mathbb{R}$, and $s \colon \mathbb{R}^d \to \mathbb{R}$ denote an arbitrary vector-valued function, activation function in FF (e.g., GELU, Hendrycks & Gimpel, 2016), element-wise mean, and element-wise standard deviation, respectively. $h$ denotes the head number in the multi-head attention. See the original paper (Vaswani et al., 2017) for more details.

**Attention map:** Our interest lies in how strongly each input $\boldsymbol{x}_j$ contributes to the computation of a particular output $\boldsymbol{y}_i$. This is typically visualized as an $n \times n$ attention map, where the $(i, j)$ cell represents how strongly $\boldsymbol{x}_j$ contributed to compute $\boldsymbol{y}_i$. A typical approximation of such a map is facilitated with the attention weights ($\alpha_{i,j}^h$ or $\sum_h \alpha_{i,j}^h$; henceforth denoted $\alpha_{i,j}$) (Clark et al., 2019; Kovaleva et al., 2019); however, this only reflects a specific process (QK attention computation) of the Transformer layer (Brunner et al., 2020; Kobayashi et al., 2020; Abnar & Zuidema, 2020).

**Looking into Transformer layer through attention map:** Nevertheless, an attention map is not only for visualizing the attention weights; existing studies (Kobayashi et al., 2020; 2021) have analyzed other components through the lens of refined attention map. Specifically, Kobayashi et al.

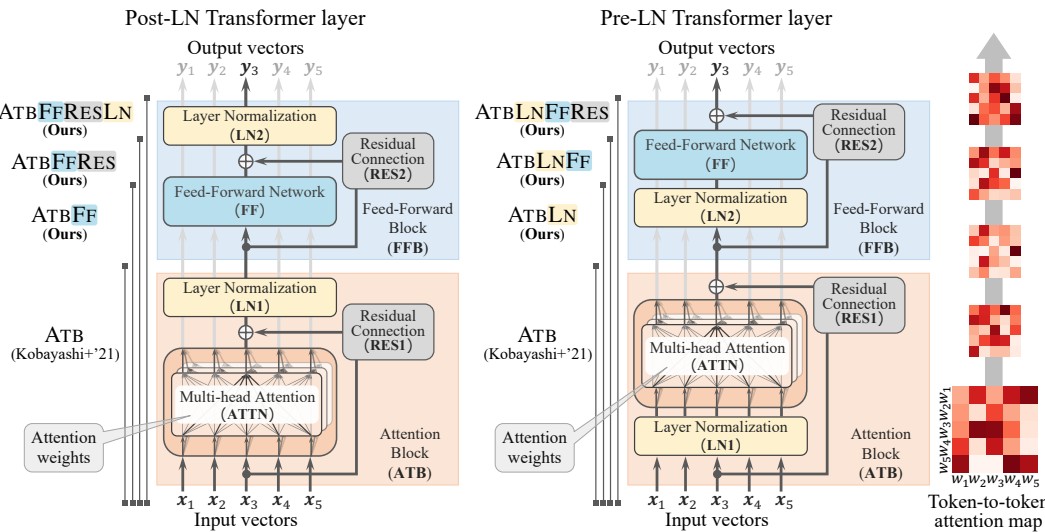

Figure 1: Overview of the Transformer layer for Post-LN and Pre-LN architectures, annotated with analysis scopes, e.g., ATBFFRESLN. The right part of this figure (token-to-token attention map) illustrates the component-by-component changes of the attention maps. See Appendix B for concrete examples of attention maps.

(2020) pointed out that the processing in the multi-head QKV attention mechanism can be written as a sum of transformed vectors:

$$\boldsymbol{y}_i^{\text{ATTN}} = \sum_j \boldsymbol{F}_i^{\text{ATTN}}(\boldsymbol{x}_j; \boldsymbol{X}) + \boldsymbol{b}^{\text{ATTN}}, \tag{6}$$

$$\text{where} \quad \boldsymbol{F}_i^{\text{ATTN}}(\boldsymbol{x}_j; \boldsymbol{X}) := \alpha_{i,j}^h(\boldsymbol{x}_j \boldsymbol{W}_V^h)\boldsymbol{W}_O^h, \quad \boldsymbol{b}^{\text{ATTN}} := \boldsymbol{b}_V \boldsymbol{W}_O + \boldsymbol{b}_O. \tag{7}$$

Here, input representation $\boldsymbol{x}_i$ is updated to $\boldsymbol{y}_i^{\text{ATTN}}$ by aggregating transformed inputs $\boldsymbol{F}_i^{\text{ATTN}}(\boldsymbol{x}_j; \boldsymbol{X})$. Then, $\|\boldsymbol{F}_i^{\text{ATTN}}(\boldsymbol{x}_j; \boldsymbol{X})\|$ instead of $\alpha_{i,j}$ alone is regarded as *refined* attention weight with the simple intuition that larger input contributes to output more in summation. This refined weight reflects not only the original attention weight $\alpha_{i,j}$, but also the effect of surrounding processing involved in $\boldsymbol{F}$, such as value vector transformation. We call this strategy of decomposing the process into the sum of transformed vectors and measuring their norms **norm-based analysis**, and the obtained $n \times n$ attribution matrix is generally called **attention map** in this study.

This norm-based analysis has been further generalized to a broader component of the Transformer layer. Specifically, Kobayashi et al. (2021) showed that the operation of attention block (ATB) could also be rewritten into the sum of transformed inputs and a bias term:

$$\boldsymbol{y}_i^{\text{ATB}} = \text{ATB}(\boldsymbol{x}_i; \boldsymbol{X}) = \sum_j \boldsymbol{F}_i^{\text{ATB}}(\boldsymbol{x}_j; \boldsymbol{X}) + \boldsymbol{b}^{\text{ATB}}. \tag{8}$$

Then, the norm $\|\boldsymbol{F}_i^{\text{ATB}}(\boldsymbol{x}_j; \boldsymbol{X})\|$ was analyzed to quantify how much the input $\boldsymbol{x}_j$ impacted the computation of the output $\boldsymbol{y}_i^{\text{ATB}}$ through the ATBs. See Appendix C for details of $\boldsymbol{F}_i^{\text{ATB}}$ and $\boldsymbol{b}^{\text{ATB}}$.

## 3 PROPOSAL: ANALYZING FFBS THROUGH REFINED ATTENTION MAP

The transformer layer is not only an ATB; it consists of an ATB and FFB (Figs. 1 and 2). Thus, this study broadens the analysis scope to include the entire FF block consisting of feed-forward, residual, and normalization layers, in addition to the ATB. Note that FFBs do not involve token-wise interaction among the inputs $\boldsymbol{X}$; thus, the layer output $\boldsymbol{y}_i^{\text{Layer}}$ can be written as transformed $\boldsymbol{y}_i^{\text{ATB}}$. Then, our aim is to decompose the entire layer processing as follows:

$$\boldsymbol{y}_i^{\text{Layer}} = \text{FFB}\left(\sum_j \boldsymbol{F}_i^{\text{ATB}}(\boldsymbol{x}_j; \boldsymbol{X}) + \boldsymbol{b}^{\text{ATB}}\right) \tag{9}$$

$$= \sum_j \boldsymbol{F}_i^{\text{Layer}}(\boldsymbol{x}_j; \boldsymbol{X}) + \boldsymbol{b}^{\text{Layer}}. \tag{10}$$

Here, the norm $\|\boldsymbol{F}_i^{\text{Layer}}(\boldsymbol{x}_j; \boldsymbol{X})\|$ can render the updated attention map reflecting the processing in the entire layer. This attention map computation can also be performed component-by-component; we can track each FFB component's effect.

**Why FFBs can control contextualization patterns:** While FFBs are applied independently to each input representation, FFB's each input already contains mixed information from multiple token representations due to the ATB's process, and the FFBs can freely modify these weights through a nonlinear transformation (see Fig. 2).

**Difficulty to incorporate FF:** While FFBs have the potential to significantly impact contextualization patterns, incorporating the FF component into norm-based analysis is challenging due to its nonlinear activation function, $\boldsymbol{g}$ (Eq. 3), which cannot be decomposed additively by the distributive law:

$$\boldsymbol{g}\Big(\textstyle\sum_j \boldsymbol{F}(\boldsymbol{x}_j)\Big) \neq \textstyle\sum_j (\boldsymbol{g} \circ \boldsymbol{F})(\boldsymbol{x}_j). \tag{11}$$

This inequality matters in the transformation from Eq. 9 to Eq. 10. That is, simply measuring the norm $\|(\boldsymbol{g} \circ \boldsymbol{F})(\boldsymbol{x}_j)\|$ is not mathematically valid. Indeed, the FF component has been excluded from previous norm-based analyses. Note that the other components in FFBs, residual connection and layer normalization, can be analyzed in the same way proposed in Kobayashi et al. (2021).

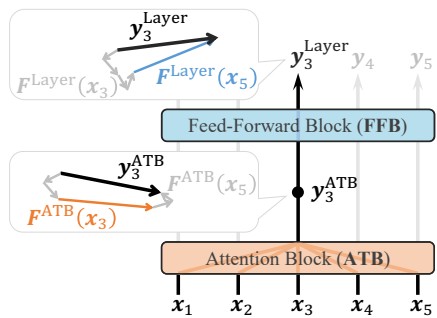

Figure 2: An illustration of possible contextualization effects by FFB. The FFB does not have the function of mixing input tokens together; however, its input already contain mixed information from multiple tokens, and the FFB is capable of altering these weights. Here, output $\boldsymbol{y}_3$ is computed based on $[\boldsymbol{x}_1, \cdots, \boldsymbol{x}_5]$. Based on the vector norm, the most influential input seems to be $\boldsymbol{x}_3$ before FF; however, after the FF's transformation, $\boldsymbol{x}_5$ becomes the most influential input.

**Integrated Gradients (IG):** The IG (Sundararajan et al., 2017) is a technique for interpreting deep learning models by using integral and gradient calculations. It measures the contribution of each input feature to the output of a neural model. Given a function $f\colon \mathbb{R}^n \to \mathbb{R}$ and certain input $\boldsymbol{x}' = (x'_1, \dots, x'_n) \in \mathbb{R}^n$, IG calculates the contribution (attribution) score $(\in \mathbb{R})$ of each input feature $x'_i \in \mathbb{R}$ to the output $f(\boldsymbol{x}') \in \mathbb{R}$:

$$f(\boldsymbol{x}') = \sum_{j=1}^n \text{IG}_j(\boldsymbol{x}'; f, \boldsymbol{b}) + f(\boldsymbol{b}), \quad \text{IG}_i(\boldsymbol{x}'; f, \boldsymbol{b}) \coloneqq (x'_i - b_i) \int_{\alpha=0}^1 \frac{\partial f}{\partial x_i}\bigg|_{\boldsymbol{x} = \boldsymbol{b} + \alpha(\boldsymbol{x}' - \boldsymbol{b})} \mathrm{d}\alpha. \tag{12}$$

Here, $\boldsymbol{b} \in \mathbb{R}^n$ denotes a baseline vector used to estimate the contribution. At least in this study, it is set to a zero vector, which makes zero output when given into the activation function ($\boldsymbol{g}(\boldsymbol{0}) = \boldsymbol{0}$), to satisfy desirable property for the decomposition (see Appendix D.2 for details).

**Expansion to FF:** We explain how to use IG for the decomposition of FF output. As aforementioned, the problem lies in the nonlinear part, thus we focus on the decomposition around the nonlinear activation. Let us define $\boldsymbol{F}_i^{\text{Pre } g}(\boldsymbol{x}_j) \coloneqq \boldsymbol{F}_i^{\text{ATB}}(\boldsymbol{x}_j; \boldsymbol{X})\boldsymbol{W}_1 \in \mathbb{R}^{d'}$ as the decomposed vector prior to the nonlinear activation $\boldsymbol{g}\colon \mathbb{R}^{d'} \to \mathbb{R}^{d'}$. The activated token representation $\boldsymbol{y}' \in \mathbb{R}^{d'}$ is written as follows:

$$\boldsymbol{y}'_i = \boldsymbol{g}\Big(\sum_{j=1}^n \boldsymbol{F}_i^{\text{Pre } g}(\boldsymbol{x}_j)\Big) = \sum_{j=1}^n \boldsymbol{h}_j\big(\boldsymbol{F}_i^{\text{Pre } g}(\boldsymbol{x}_1), \dots, \boldsymbol{F}_i^{\text{Pre } g}(\boldsymbol{x}_n); \widetilde{g}, \boldsymbol{0}\big), \tag{13}$$

$$\boldsymbol{h}_j\big(\boldsymbol{F}_i^{\text{Pre } g}(\boldsymbol{x}_1), \dots, \boldsymbol{F}_i^{\text{Pre } g}(\boldsymbol{x}_n); \widetilde{g}, \boldsymbol{0}\big) = \begin{bmatrix} \text{IG}_j\big(\boldsymbol{F}_i^{\text{Pre } g}(\boldsymbol{x}_1)[1], \dots, \boldsymbol{F}_i^{\text{Pre } g}(\boldsymbol{x}_n)[1]; \widetilde{g}, \boldsymbol{0}\big) \\ \vdots \\ \text{IG}_j\big(\boldsymbol{F}_i^{\text{Pre } g}(\boldsymbol{x}_1)[d'], \dots, \boldsymbol{F}_i^{\text{Pre } g}(\boldsymbol{x}_n)[d']; \widetilde{g}, \boldsymbol{0}\big) \end{bmatrix}^\top, \tag{14}$$

where the transformation $\widetilde{g}\colon \mathbb{R}^n \to \mathbb{R}$ is defined as $\widetilde{g}(x_1, \dots, x_n) \coloneqq g(x_1 + \cdots + x_n)$, which adds up the inputs into a single scalar and then applies the element-level activation $g\colon \mathbb{R} \to \mathbb{R}$. The function $\boldsymbol{h}_j\colon \mathbb{R}^{n \times d'} \to \mathbb{R}^{d'}$ yields how strongly a particular input $\boldsymbol{F}_i^{\text{Pre } g}(\boldsymbol{x}_j)$ contributed to the output $\boldsymbol{y}'_i$. Each element $\boldsymbol{h}_j[k]$ indicates the contribution of the $k$-th element of input $\boldsymbol{F}_i^{\text{Pre } g}(\boldsymbol{x}_j)[k]$

to the $k$-th element of output $\boldsymbol{y}_i'[k]$. Note that contribution can be calculated element-level because the activation $g$ is applied independently to each element.

Notably, the sum of contributions across inputs matches the output (Eq. 13), achieved by the desirable property of IG—*completeness* Sundararajan et al. (2017). This should be satisfied in the norm-based analysis and ensures that the output vector and the sum of decomposed vectors are the same. The norm $\|\boldsymbol{h}_j(\boldsymbol{F}_i^{\text{Pre } g}(\boldsymbol{x}_1), \ldots, \boldsymbol{F}_i^{\text{Pre } g}(\boldsymbol{x}_n); \widetilde{g}, \boldsymbol{0})\|$ is interpreted as the contribution of input $\boldsymbol{x}_j$ to the output $\boldsymbol{y}_i'$ of the activation in FF.

**Expansion to entire layer:** Expanding on this, the contribution of a layer input $\boldsymbol{x}_j$ to the $i$-th FF output is exactly calculated as $\|\boldsymbol{h}_j(\boldsymbol{F}_i^{\text{Pre } g}(\boldsymbol{x}_1), \ldots, \boldsymbol{F}_i^{\text{Pre } g}(\boldsymbol{x}_n); \widetilde{g}, \boldsymbol{0})\boldsymbol{W}_2\|$. Then, combined with the exact decomposition of ATTN, RES, and LN shown by Kobayashi et al. (2020; 2021), the entire Transformer layer can be written as the sum of vector-valued functions with each input vector $\boldsymbol{x}_j \in \boldsymbol{X}$ as an argument as written in Eq. 10. This allows us to render the updated attention map reflecting the processing in the entire layer by measuring the norm of each decomposed vector.

## 4 GENERAL EXPERIMENTAL SETTINGS

**Estimating refined attention map:** To elucidate the input contextualization effect of each component in FFBs, we computed attention maps by each of the following four scopes (Fig. 1):

- ATB (Kobayashi et al., 2021): Analyzing the attention block (i.e., ATTN, RES1, and LN1) using vector norms as introduced in Eq. 8.
- ATBFF (**proposed**): Analyzing components up to FF using vector norms and IG.
- ATBFFRES (**proposed**): Analyzing components up to RES2 using vector norms and IG.
- ATBFFRESLN (**proposed**): Analyzing the whole layer (all components) using vector norms and IG.

We will compare the attention maps from different scopes to separately analyze the contextualization effect. Note that if the model adopts the Pre-LN architecture, the scopes will be expanded from ATB to ATBLN, ATBLNFF, and ATBLNFFRES (see the Pre-LN part in Fig. 1).

**Models:** We analyzed 11 variants of the masked language models: six BERTs (uncased) with different sizes (large, base, medium, small, mini, and tiny) (Devlin et al., 2019; Turc et al., 2019), three BERTs-base with different seeds (Sellam et al., 2022), plus two RoBERTas with different sizes (large and base) (Liu et al., 2019). We also analyzed two causal language models: GPT-2 with 117M parameters and OPT (Zhang et al., 2022) with 125M parameters. Note that the masked language models adopt the Post-LN architecture, and the causal language models adopts the Pre-LN architecture.

**Data:** We used two datasets with different domains: Wikipedia excerpts (992 sequences) (Clark et al., 2019)[1] and the Stanford Sentiment Treebank v2 dataset (872 sequences from validation set) (SST-2, Socher et al., 2013). The input was segmented by each model's pre-trained tokenizers; analysis was conducted at the subword level.[2]

## 5 EXPERIMENT 1: CONTEXTUALIZATION CHANGE

Does each component in FFBs indeed modify the token-to-token contextualization? We analyze the contextualization change through each component in FFBs.

### 5.1 MACRO CONTEXTUALIZATION CHANGE

**Calculating contextualization change:** Given two analysis scopes (e.g., before and after FF; ATB $\leftrightarrow$ ATBFF), their contextualization pattern change was quantified following some procedures of representational similarity analysis (Kriegeskorte et al., 2008). Formally, given an input sequence of length $n$, two different attention maps from the two scopes are obtained. Then, each attention map ($\mathbb{R}^{n \times n}$) was flattened into a vector ($\mathbb{R}^{n^2}$), and the Spearman's rank correlation coefficient $\rho$ between

---

[1]https://github.com/clarkkev/attention-analysis
[2]For masked language models, each sequence was fed into the models with masking 12% tokens as done in the case of the BERT training procedure.

the two vectors were calculated. We report the average contextualization change $1 - \rho$ across input sequences. We will report the results of the BERT-base and GPT-2 on the Wikipedia dataset in this section; other models (including OPT) and datasets also yielded similar results (see Appendix G.1).

The contextualization changes through each component in FFBs are shown in Fig. 3. A higher score indicates that the targeted component more drastically updates the contextualization patterns. Note that we explicitly distinguish the pre- and post-layer normalization (PRELN and POSTLN) in this section, and the component order in Fig. 3 is the same as the corresponding layer architecture.

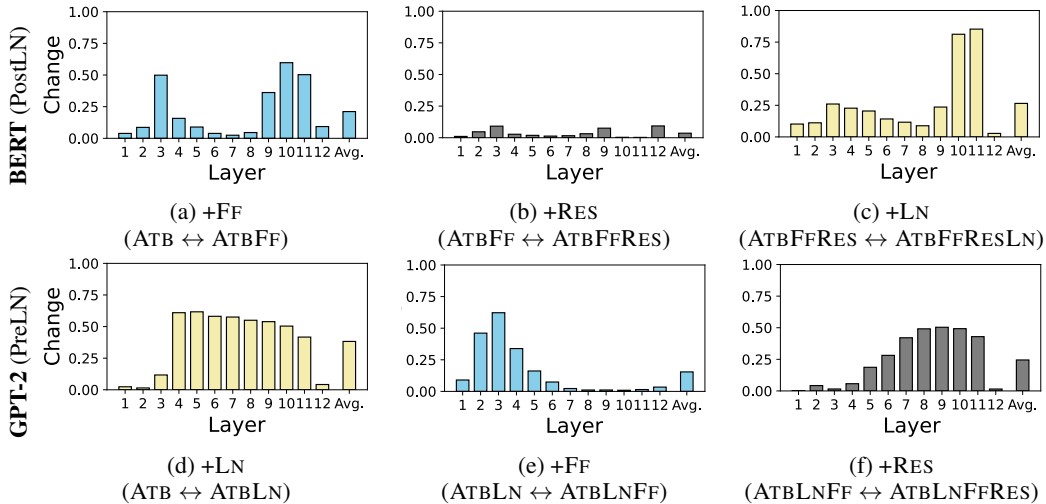

Figure 3: Contextualization changes between before and after each component in FFBs (FF, RES2, and LN2) of BERT and GPT-2. The higher the bar, the more drastically the token-to-token contextualization (attention maps) changes due to the target component.

We generally observed that each component did modify the input contextualization, especially in particular layers. For example, in the BERT-base, the FF in 3rd and 9th–11th layers and normalization in 10th–11th layers incurred relatively large contextualization change. Comparing the BERT-base and GPT-2, the by-layer average of contextualization change by FF and LN was similar across these LMs: $0.21$ and $0.15$ for FF in BERT and GPT-2 and $0.27$ and $0.38$ for LN in BERT and GPT-2, respectively. In contrast, there seem mainly two differences between BERT and GPT-2: (i) the effect of RES, and (ii) the layers at which the high-impact FFs are located. At least for the FFs of GPT-2, the shallow layer's high impact is consistent with some existing studies (Meng et al., 2022; Geva et al., 2023; Gromov et al., 2024), but revealing the cause of such gaps between BERT and GPT-2 would be future work. Note that Figs. 14 to 18 in Appendix G show the results for other model variants, yielding somewhat consistent trends that FFs and LNs in particular layers especially incur contextualization changes.

## 5.2 LINGUISTIC PATTERNS IN FF'S CONTEXTUALIZATION EFFECTS

The FF network is a completely new scope in the norm-based analysis, while residual and normalization layers have been analyzed in Kobayashi et al. (2021); Modarressi et al. (2022). Thus, we further investigate how FF modified contextualization, using BERT-base and GPT-2 on the Wikipedia dataset as a case study.

**Micro contextualization change:** We compared the attention maps before and after FF. Specifically, we subtract a pre-FF attention map from a post-FF map (Fig. 4); we call the resulting diff-map **FF-amp matrix** and the values in each cell **FF-amp score**.[3] A larger FF-amp score of the $(i, j)$ cell

---

[3] Before the subtraction, the two maps were normalized so that the sum of the values of each column was 1; this normalization facilitates the inter-method comparison.

presents that the contribution of input $x_j$ to output $y_i$ is more amplified by FF. Our question is which kind of token pairs gain a higher FF-amp score.[4]

**FFs amplify particular linguistic compositions:** Based on the preliminary observations of high FF-amp token pairs, we set 7 linguistic categories typically amplified by FFBs. This includes, for example, token pairs consisting of the same words but different positions in the input. Table 1 shows the top two token pairs w.r.t. the amp score

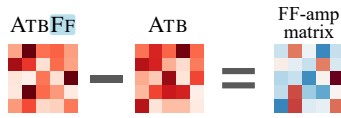

Figure 4: FF-amp matrix is computed by subtracting the attention map before FF (ATB) from that after FF (ATBFF).

in some layers and their token pair category. Fig. 5 summarizes the ratio of each category in the top 50 token pairs with the highest FF-amp score.[5] Compared to the categories for randomly sampled token pairs (FR; the second rightmost bar in Fig. 5) and adjacent token pairs (AR; the rightmost bar in Fig. 5), the amplified pairs in each layer have unique characteristics; for example, in the former layers, subword pairs consisting the same token are highly contextualized by FFs. Note that this phenomenon of processing somewhat shallow morphological information in the former layers can be consistent with the view of BERT as a pipeline perspective, with gradually more advanced processing from the former to the latter layers Tenney et al. (2019).

Table 1: Token pairs sampled from the top 10 most-amplified subword pairs by FF in each layer of BERT (left) and GPT-2 (right). See Tables 2 and 3 in Appendix G.2 for the full pairs. The text colors are aligned with word pair categories, and the same colors are used in Fig. 5.

| Layer | amplified pairs by BERT's FF | Layer | amplified pairs by GPT-2's FF | |
|---|---|---|---|---|
| 1 | (opera, soap), (##night, week) | 1 | (ies, stud), (ning, begin) | Subword |
| 3 | (toys, ##hop), (' , t) | 3 | (_than, _rather), (_others, _among) | Compound noun |
| 6 | (with, charged), (fleming, colin) | 6 | (_del, del), (_route, route) | Common expression |
| 9 | (but, difficulty), (she, teacher) | 8 | (_08, _2007), (_14, _density) | Same token |
| 11 | (tiny, tiny), (highway, highway) | 12 | (_operational, _not), (_daring, _has) | Named entity |
| | | | | Semantical connection |
| | | | | Others |

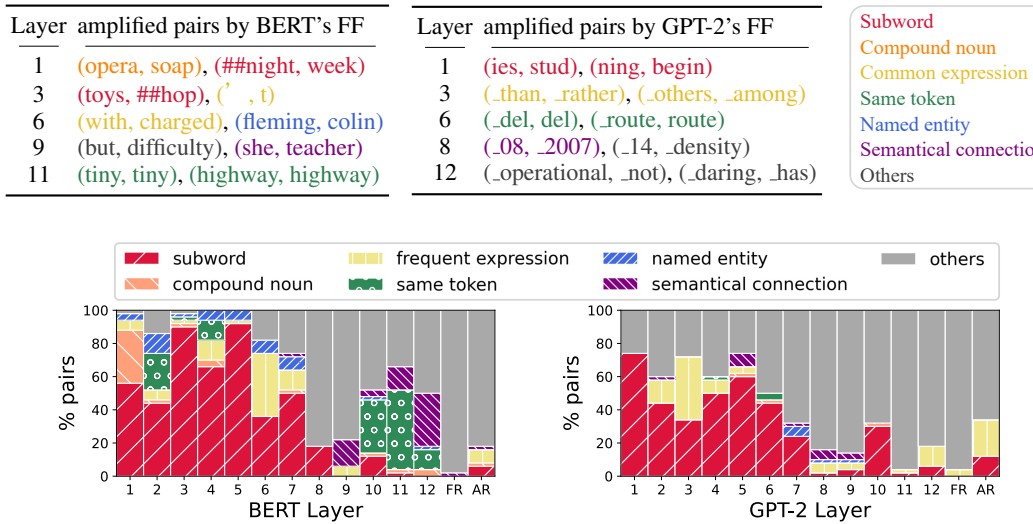

Figure 5: Breakdown of the category labels we manually assigned to top 50 pairs having the largest FF-amp score in each layer of BERT and GPT-2. We also assigned the labels to fully random 50 pairs ("FR") and adjacent random 50 pairs ("AR").

**Simple word co-occurrence does not explain the FF's amplification:** Do FFs simply amplify the interactions between frequently co-occurring words? We additionally investigated the relationship between the FF's amplification and word co-occurrence. Specifically, we calculated PMI for each subword pair on the Wikipedia dump and then calculated the Spearman's rank correlation coefficient between FF-amp scores of each pair $(w_i, w_j)$ from BERT-base and PMI values.[6] We observed that the correlation scores were fairly low in any layer (coefficient values were 0.06–0.14). Thus, we found that the FF does not simply modify the contextualization based on the word co-occurrence.

---

[4]We aggregated the average FF-amp score for each subword type pair $(w_i, w_j)$. Pairs consisting of the same position's token $(w_i, w_i)$ and pairs occurring only once in the dataset were excluded.

[5]One of the authors of this paper has conducted the annotation.

[6]We defined three types of co-occurrences in calculating PMI: the simultaneous occurrence of two subwords (i) in an article, (ii) in a sentence, and (iii) in a chunk of 512 tokens. No correlation was observed for either type of co-occurrence. Pairs including special tokens and pairs consisting of the same subword were excluded.

To sum up, these findings indicate that FF did amplify the contextualization aligned with particular types of linguistic compositions in various granularity (i.e., word and NE phrase levels).

# 6 EXPERIMENT 2: DYNAMICS OF CONTEXTUALIZATION CHANGE

We analyze the relationship between the contextualization performed by FF and other components, such as RES, given the previous observation that contextualization changes in a particular component are sometimes overwritten by other components (Kobayashi et al., 2021). We will report the results of the BERT-base and GPT-2 on the Wikipedia dataset in this section; other models/datasets also yielded similar results (see Appendix H).

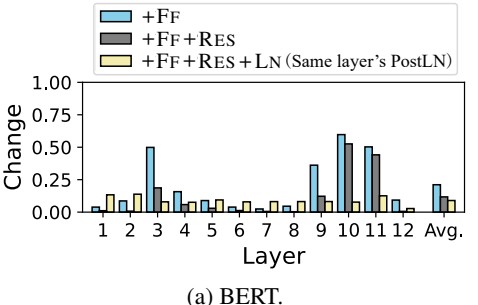 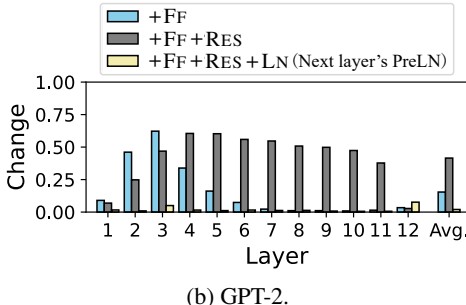

(a) BERT.        (b) GPT-2.

Figure 6: Contextualization changes through FF, RES, and LN relative to the contextualization performed before FF. The higher the bar, the more the contextualization changes.

Fig. 6 shows the contextualization change scores ($1 - \rho$ described in § 5.1) by FF and subsequent components: FF (+FF), FF and RES (FF+RES), and FF, RES and LN (FF+RES+LN). Note that we analyzed the next-layer's LN1 in the case of GPT2 (Pre-LN architecture) to analyze the BERTs and GPTs from the same perspective—*whether the other components overwrite the contextualization performed in the FF network*. If a score is zero, the resulting contextualization map is the same as that computed before FF (after ATBs in BERT or after LN2 in GPT-2). The notable point is that through the FF and subsequent RES and LN, the score once becomes large but finally converges to be small; that is, the contextualization by FFs tends to be canceled by the following components. We look into this cancellation phenomenon with a specific focus on each component.

## 6.1 FF AND RES

The residual connection bypasses the feed-forward layer as Eq. 4. Here, the interest lies in how dominant the bypassed representation $x'_i$ is relative to $\mathrm{FF}(x'_i)$. For example, if the representation $x'_i$ has a much larger L2 norm than $\mathrm{FF}(x'_i)$, the final output through the $\mathrm{RES2} \circ \mathrm{FF}$ will be similar to the original input $x'_i$; that is, the contextualization change performed in the FF would be diminished.

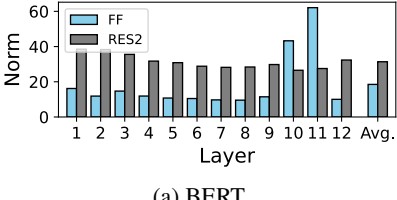 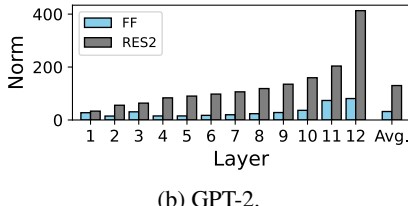

(a) BERT.        (b) GPT-2.

Figure 7: Averaged norm of the output vectors from FF and the bypassed vectors via RES2, calculated on the Wikipedia data for BERT and GPT-2.

**RES2 adds a large vector:** We observe that the vectors bypassed via RES2 are more than twice as large as output vectors from FF in the L2 norm in most layers (Fig. 7). That is, the representation (contextualization) updated by the FF tends to be overwritten/canceled by the original one. This observation is consistent with that of RES1 (Kobayashi et al., 2021). Notably, this cancellation is

weakened in the 10th–11th layers in BERT and former layers in GPT2s, where FFs' contextualization was relatively large (Figs. 3a and 3e).

## 6.2 FF AND LN

We also analyzed the relationship between the contextualization performed in FF and LN. Note that the layer normalization first normalizes the input representation, then multiplies a weight vector $\gamma$ element-wise, and adds a bias vector $\beta$ (Eq. 8).

**Cancellation mechanism:** Again, as shown in Fig. 6, the contextualization change after the LN (+FF+RES+LN) is much lower than in preceding scopes (+FF and +FF+RES). That is, the LNs substantially canceled out the contextualization performed in the preceding components of FF and RES. Then, we specifically tackle the question, how did LN cancel the FF's effect?

We first found that the FF output representation has outliers in some specific dimensions (green lines in Fig. 8), and that the weight $\gamma$ of LN tends to shrink these special dimensions (red lines in Fig. 8). In the layers where FF incurs a relatively large impact on contextualization,[7] the Pearson correlation coefficient between LN's $\gamma$ and mean absolute value of FF output by dimension was from $-0.45$ to $-0.78$ in BERTs and from $-0.22$ to $-0.59$ in GPT-2 across layers. Thus, we suspect that such specific outlier dimensions in the FF outputs encoded "flags" for potential contextualization change, and LN typically declines such FF's suggestions by erasing the outliers.

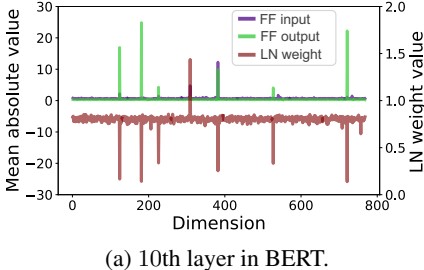
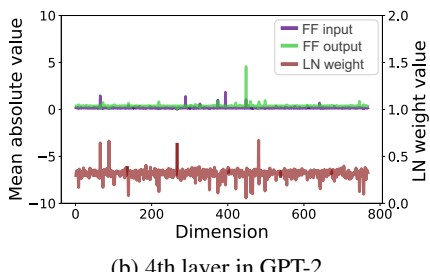

(a) 10th layer in BERT.  (b) 4th layer in GPT-2.

Figure 8: Mean absolute value in each dimension of the input/output vectors of FF across the Wikipedia data and the LN weight values at the certain layer.

Indeed, we observed that ignoring such special outlier dimensions (bottom $1\%$ with the lowest value of $\gamma$) in calculating FF's contextualization makes the change score quite small; contextualization changes by FF went from 0.21 to 0.09 in BERT and from 0.15 to 0.02 in GPT-2 on a layer average. Thus, FF's contextualization effect is realized using very specific dimensions, and LN2 cancels the effect by shrinking these special dimensions. Note that a related phenomenon was discovered by Modarressi et al. (2022): LN2 and LN1 cancel each other out by outliers in their weights in BERT (see other related works in A.3). The observed cancellation mechanism over feed-forward and normalization layers also suggests redundant contextualization processing in Transformer (A.4). Further investigation is an important future work.

## 7 CONCLUSIONS AND FUTURE WORK

We have analyzed the FF blocks w.r.t. input contextualization through the lens of a refined attention map by leveraging the existing norm-based analysis and the integrated gradient method having an ideal property—completeness. Our experiments using masked- and causal-language models have shown that FFs indeed modify the input contextualization by amplifying specific types of linguistic compositions (e.g., subword pairs forming one word). We have also found that FF and its surrounding components tend to cancel out each other's contextualization effects and clarified their mechanism, implying the redundancy of the processing within the Transformer layer. Applying our analysis to other model variants, such as Mistral (model with local attention, Jiang et al., 2023) and Mixtral (model with mixture of experts, Jiang et al., 2024) will be our future work. In addition, focusing on inter-layer contextualization dynamics could also be fascinating future directions.

---

[7]FFs in BERT's 3rd and 10th–11th layers and GPT-2's 1st–11th layers

## ETHICS STATEMENT

We recognize that this study has not used any special data or methods with potential ethical issues. One recent issue in the whole ML community is that neural-network-based models make predictions containing non-intended *biases* (e.g., gender bias). This paper gives a method for interpreting the inner workings of real-world machine learning models, which may help us understand such biased behaviors of the models in the future.

## REPRODUCIBILITY

We have reported the used data and pre-trained models in the manuscript as much as possible, and no private or unofficial data people can not access are used. In addition, we used several variants of models, including different types of masked- and causa-LMs; this hopefully has contributed to enhancing our results' generality (reproducibility). All the models can be publicly available through Huggingface's transformers framework (Wolf et al., 2020). Additionally, our analysis tool and experimental codes are publicly available at `https://github.com/gorokoba560/norm-analysis-of-transformer`.

## ACKNOWLEDGMENTS

We would like to thank the members of the Tohoku NLP Group for their insightful comments. This work was supported by JSPS KAKENHI Grant Number JP22J21492, JP22H05106; JST CREST Grant Number JPMJCR20D2, Japan; and JST ACT-X Grant Number JPMJAX200S, Japan.

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

## A    RELATED WORKS

This section supplements the studies directly related to the methods or experimental results of this paper, which are not fully discussed in the main body.

### A.1    MECHANISTIC INTERPRETABILITY FOR TRANSFORMER LMS

> *Mechanistic interpretability seeks to reverse engineer neural networks, similar to how one might reverse engineer a compiled binary computer program.* (Olah, 2022)

In the past decade, as deep learning has become the predominant approach for classification and representation learning, interpreting and understanding high-dimensional, nonlinear language models has emerged as a significant challenge for the NLP community. The NLP and closely related machine learning communities have amassed a vast array of scientific and engineering insights into the interpretation of deep learning models under the banners of interpretability, explainability, XAI, and probing. Meanwhile, in a diverging line of research, researchers from organizations such as Google, Anthropic, and OpenAI have published compelling research under the banner of mechanistic interpretability, focusing on the interpretation of internal parameters of deep learning models. Although these two approaches share many technical commonalities, the exchange between them is not yet as active as it could be. In this section, we will overview the trend of mechanistic interpretability for the NLP/ML community and specifically highlight research that is closely related to this paper.

Mechanistic interpretability begins with the analysis of toy or small-scale language models, aiming for a thorough and comprehensive understanding of internal phenomena and mechanisms, interpreting parameters and activations, and providing tools for the analysis. The interest in mechanistic interpretability for Transformer language models (LMs) has been growing, particularly with Anthropic's Transformer Circuits Thread[8]. Our study, the analysis of attention maps (mixing between tokens), focuses on how information from specific tokens propagates to surrounding tokens in the model. Therefore, our study may be seen as operations similar to the assembler's MOV and ADD commands from the perspective of mechanistic interpretability.

Additionally, the attention map is used as an analytical tool in mechanistic interpretability. Elhage et al. (2022c) discovered specific attention heads (called Induction heads) through the observation of attention maps calculated with norm-based analysis. Catherine Olsson et al. (2022) revealed that the induction heads contribute to the in-context learning of large-scale language models.

In mechanistic interpretability, several studies have attempt to interpret FF (MLP) layers, but challenges in understanding and interpretation have been raised (Elhage et al., 2022b;a). We believe that our method or results for understanding FF through the lens of attention maps can be beneficial to the field of mechanistic interpretability.

---

[8]`https://transformer-circuits.pub/`

A.2   ATTENTION MAP (ANALYSIS OF MIXING BETWEEN TOKENS)

The core component of the Transformer is the self-Attention mechanism, and focusing on the interactions between tokens (attention maps) has become one of the mainstream analytical approaches. The computation of attention maps starts with attention weights and has been improved or extended (Clark et al., 2019; Kovaleva et al., 2019; Brunner et al., 2020; Kobayashi et al., 2020; 2021).

One of the key extension strategies is the **norm-based analysis**, which can consider the effects of all surrounding matrices, vectors, and modules without approximation, and also satisfies the axiom of the additive feature attribution methods (Lundberg & Lee, 2017). Initially, Kobayashi et al. (2020) proposed a norm-based computation method that considers the entire attention block, ATTN; subsequently, Kobayashi et al. (2021) expanded this approach to include the effects of the RES1 and LN1. For technical details, please refer to § 2. The norm-based analysis provides insights into the internal mechanisms of models; for example, attention maps computed with norm-based analysis are used to measure similarities between language models and human behavior, e.g., reading time (Oh & Schuler, 2022).

This paper proposes and analyzes a method for calculating attention maps across an entire layer. However, as related research or future work, it is also possible to calculate attention maps across the entire Transformer architecture. Abnar & Zuidema (2020) computed an attention map at each layer by adding an identity matrix to the attention weights matrix (accounting for ATTN and RES1), and then integrated these maps (all the layers) in two ways: (i) Attention rollout uses recursive matrix multiplication and (ii) Attention flow uses a maximum flow algorithm. Ferrando et al. (2022) proposed ALTI, which uses norm-based analysis to compute attention maps at each layer considering the Attention block (ATTN, RES1, LN1), and integrates them using Attention rollout. Modarressi et al. (2022) proposed GlobEnc, which uses norm-based analysis to compute Attention maps at each layer considering all modules except FF (i.e., ATTN, RES1, LN1, RES2, and LN2), and integrates them using Attention rollout. Modarressi et al. (2023) proposed DecompX, which decomposes the processes of the entire model similarly to the norm-based analysis and calculates an attention map for the entire model relative to the model's prediction (logit).

DecompX (Modarressi et al., 2023) is the only method among those previously mentioned that considers the nonlinear activation functions in the FF layers. Therefore, from this perspective, DecompX can be considered a competitor to our proposed approach. We will discuss the differences in greater detail in the following. DecompX uses a coarse linear approximation for decomposing the non-linear function in FF. Our study differs from DecompX in two aspects: (i) the scope of the study (visualization tool for model-wide behavior vs. analysis of module function) and (ii) the decomposition of FF (with vs. without approximation). Investigating the impact of the approximation in the decomposition of FF on our analysis is an important future work.

A.3   OUTLIERS IN TRANSFORMERS

§ 6.2 demonstrated that FF and LN2 exhibit unique behaviors concerning outliers. It is well-known that the internal representations of Transformer LMs tend to have outliers (Luo et al., 2021; Kovaleva et al., 2021). It has also been revealed that FF generates outliers in certain dimensions (Ferrando et al., 2023; Bondarenko et al., 2023), and that the parameters of LN contains outliers (Modarressi et al., 2022; Puccetti et al., 2022). However, to our knowledge, our finding that FF and LN2 cancel out each other through these outliers is novel and suggests significant insights into the redundancy of the model.

A.4   REDUNDANCY OF TRANSFORMERS

As mentioned above, our results (§ 6.2) suggest redundancy in the Transformer architecture. Redundancy of Transformer has also been uncovered from the success of parameter reduction. Successful pruning has been achieved for the ATTN (Michel et al., 2019; Kovaleva et al., 2019), FF (Santacroce et al., 2023), and both of them (Gromov et al., 2024; Tao et al., 2023). Additionally, attempts have been made to reduce learned parameters through low-rank approximations (Sharma et al., 2023) and to construct lightweight models using knowledge distillation (Sanh et al., 2020; Xu et al., 2024).

# B    CONCRETE EXAMPLES OF OUR ANALYSIS

Figs. 9 and 10 show concrete examples of the attention maps calculated using our analysis method for BERT and GPT-2. Fig. 9 shows that FF in the 12th layer of BERT amplifies the mixing from "Tokyo" to "`[MASK]`" for the input "`[CLS]` Tokyo is the capital of `[MASK]` . `[SEP]`". Fig. 10 shows that FF in the 3rd layer of GPT-2 amplifies the mixing from "rather" to "than" for the input "She loves math rather than engaging in".

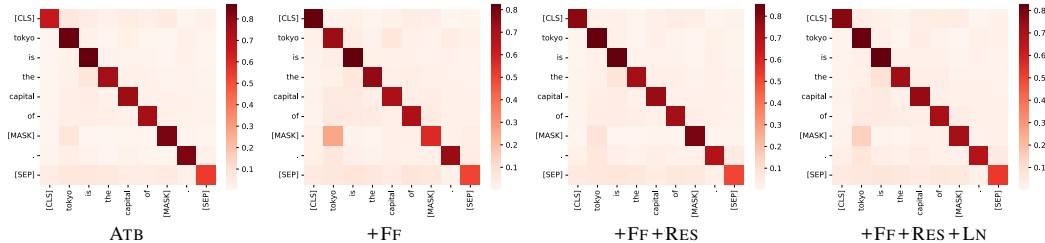

Figure 9: Attention maps computed with our analysis method for 12th layer in BERT-base.

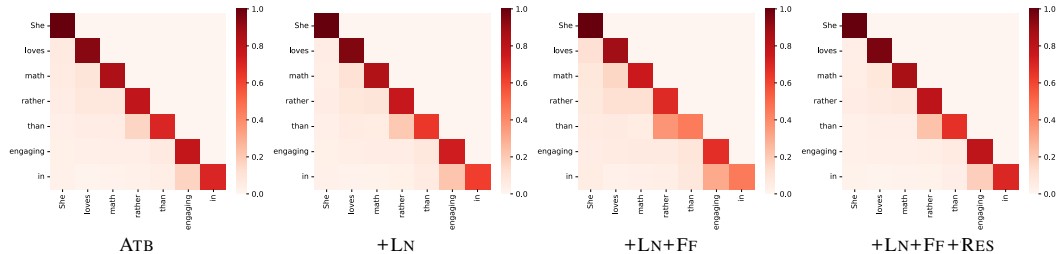

Figure 10: Attention maps computed with our analysis method for 3rd layer in GPT-2 small.

# C    DETAILED FORMULAS FOR EACH ANALYSIS METHOD

We describe the mathematical details of the norm-based analysis methods adopted in this paper.

## C.1    ATB

As described in § 2, Kobayashi et al. (2021) rewrite the operation of the attention block (ATB) into the sum of transformed inputs and a bias term:

$$\boldsymbol{y}_i^{\text{ATB}} = \text{ATB}(\boldsymbol{x}_i; \boldsymbol{X}) \tag{15}$$

$$= \text{LN1} \circ \text{RES1} \circ \text{ATTN} \tag{16}$$

$$= \sum_{j=1}^{n} \boldsymbol{F}_i^{\text{ATB}}(\boldsymbol{x}_j; \boldsymbol{X}) + \boldsymbol{b}^{\text{ATB}}. \tag{17}$$

First, the multi-head attention mechanism (ATTN; Eq. 1) can be decomposed into a sum of transformed vectors as Eq. 6 and Eq. 7:

$$\text{ATTN}(\boldsymbol{x}_i; \boldsymbol{X}) = \sum_j \sum_h \alpha_{i,j}^h (\boldsymbol{x}_j \boldsymbol{W}_V^h) \boldsymbol{W}_O^h + \boldsymbol{b}_V \boldsymbol{W}_O + \boldsymbol{b}_O \tag{18}$$

$$= \sum_j \boldsymbol{F}_i^{\text{ATTN}}(\boldsymbol{x}_j; \boldsymbol{X}) + \boldsymbol{b}^{\text{ATTN}} \tag{19}$$

$$\boldsymbol{F}_i^{\text{ATTN}}(\boldsymbol{x}_j; \boldsymbol{X}) = \alpha_{i,j}^h (\boldsymbol{x}_j \boldsymbol{W}_V^h) \boldsymbol{W}_O^h \tag{20}$$

$$\boldsymbol{b}^{\text{ATTN}} = \boldsymbol{b}_V \boldsymbol{W}_O + \boldsymbol{b}_O. \tag{21}$$

Second, the residual connection (RES; Eq. 4) just performs the vector addition. So, ATTN and RES also can be decomposed into a sum of transformed vectors collectively:

$$(\text{RES1} \circ \text{ATTN})(\boldsymbol{x}_i; \boldsymbol{X}) = \text{ATTN}(\boldsymbol{x}_i; \boldsymbol{X}) + \boldsymbol{x}_i \tag{22}$$

$$= \sum_{j \neq i} \boldsymbol{F}_i^{\text{ATTN}}(\boldsymbol{x}_j; \boldsymbol{X}) + \boldsymbol{x}_i + \boldsymbol{b}^{\text{ATTN}} \tag{23}$$

$$= \sum_j \boldsymbol{F}_i^{\text{ATTN+RES}}(\boldsymbol{x}_j; \boldsymbol{X}) + \boldsymbol{b}^{\text{ATTN}} \tag{24}$$

$$\boldsymbol{F}_i^{\text{ATTN+RES}}(\boldsymbol{x}_j; \boldsymbol{X}) = \left\{ \begin{array}{ll} \boldsymbol{F}_i^{\text{ATTN}}(\boldsymbol{x}_j; \boldsymbol{X}) & (j \neq j) \\ \boldsymbol{F}_i^{\text{ATTN}}(\boldsymbol{x}_i; \boldsymbol{X}) + \boldsymbol{x}_i & (j = i) \end{array} \right. . \tag{25}$$

Third, the layer normalization (LN; Eq. 5) performs a normalization and element-wise affine transformation. Suppose the input to LN is a sum of vectors $\boldsymbol{z} = \sum_j \boldsymbol{z}_j$, LN's operation is decomposed as follows:

$$\text{LN}(\boldsymbol{z}) = \frac{\boldsymbol{z} - m(\boldsymbol{z})}{s(\boldsymbol{z})} \odot \boldsymbol{\gamma} + \boldsymbol{\beta} \tag{26}$$

$$= \frac{\boldsymbol{z} - \frac{1}{d}\sum_{k=1}^d \boldsymbol{z}^{(k)}}{s(\boldsymbol{z})} \odot \boldsymbol{\gamma} + \boldsymbol{\beta} \tag{27}$$

$$= \frac{\sum_j \boldsymbol{z}_j - \frac{1}{d}\sum_{k=1}^d \left(\sum_j \boldsymbol{z}_j\right)^{(k)}}{s(\boldsymbol{z})} \odot \boldsymbol{\gamma} + \boldsymbol{\beta} \tag{28}$$

$$= \sum_j \frac{\boldsymbol{z}_j - \frac{1}{d}\sum_{k=1}^d \boldsymbol{z}_j^{(k)}}{s(\boldsymbol{z})} \odot \boldsymbol{\gamma} + \boldsymbol{\beta} \tag{29}$$

$$= \sum_j \frac{\boldsymbol{z}_j - m(\boldsymbol{z}_j)}{s(\boldsymbol{z})} \odot \boldsymbol{\gamma} + \boldsymbol{\beta} \tag{30}$$

$$= \sum_j \boldsymbol{F}^{\text{LN}}(\boldsymbol{z}_j) + \boldsymbol{\beta} \tag{31}$$

$$\boldsymbol{F}^{\text{LN}}(\boldsymbol{z}_j) = \frac{\boldsymbol{z}_j - m(\boldsymbol{z}_j)}{s(\boldsymbol{z})} \odot \boldsymbol{\gamma}, \tag{32}$$

where $\boldsymbol{z}^{(k)}$ denotes the $k$-th element of vector $\boldsymbol{z}$. By exploiting this decomposition, attention block (ATB; ATTN, RES, and LN) can be decomposed into a sum of transformer vectors:

$$(\text{LN1} \circ \text{RES1} \circ \text{ATTN})(\boldsymbol{x}_i; \boldsymbol{X}) = \text{LN}\left(\sum_j \boldsymbol{F}_i^{\text{ATTN+RES}}(\boldsymbol{x}_j; \boldsymbol{X}) + \boldsymbol{b}^{\text{ATTN}}\right) \tag{33}$$

$$= \sum_j \boldsymbol{F}^{\text{LN}}\left(\boldsymbol{F}_i^{\text{ATTN+RES}}(\boldsymbol{x}_j; \boldsymbol{X})\right) + \boldsymbol{F}^{\text{LN}}\left(\boldsymbol{b}^{\text{ATTN}}\right) + \boldsymbol{\beta} \tag{34}$$

$$= \sum_j \boldsymbol{F}_i^{\text{ATB}}(\boldsymbol{x}_j; \boldsymbol{X}) + \boldsymbol{b}^{\text{ATB}} \tag{35}$$

$$\boldsymbol{F}_i^{\text{ATB}}(\boldsymbol{x}_j; \boldsymbol{X}) = \boldsymbol{F}^{\text{LN}}\left(\boldsymbol{F}_i^{\text{ATTN+RES}}(\boldsymbol{x}_j; \boldsymbol{X})\right) \tag{36}$$

$$\boldsymbol{b}^{\text{ATB}} = \boldsymbol{F}^{\text{LN}}\left(\boldsymbol{b}^{\text{ATTN}}\right) + \boldsymbol{\beta}. \tag{37}$$

Then, the ATB method quantifies how much the input $\boldsymbol{x}_j$ impacted the computation of the output $\boldsymbol{y}_i^{\text{ATB}}$ by the norm $\|\boldsymbol{F}_i^{\text{ATB}}(\boldsymbol{x}_j; \boldsymbol{X})\|$.

## C.2 ATBFF

The feed-forward network (FF) via a two-layered fully connected network (Eq. 3). By exploiting the decomposition of an activation function $\boldsymbol{g}$ (§ 3), ATB and FF can be decomposed into a sum of

transformed vectors collectively:

$$(\mathrm{FF} \circ \mathrm{ATB})(\boldsymbol{x}_i; \boldsymbol{X}) = \boldsymbol{g}\left(\mathrm{ATB}(\boldsymbol{x}_i; \boldsymbol{X})\boldsymbol{W}_1 + \boldsymbol{b}_1\right)\boldsymbol{W}_2 + \boldsymbol{b}_2 \tag{38}$$

$$= \boldsymbol{g}\left(\left(\sum_{j=1}^{n} \boldsymbol{F}_i^{\mathrm{ATB}}(\boldsymbol{x}_j; \boldsymbol{X}) + \boldsymbol{b}^{\mathrm{ATB}}\right)\boldsymbol{W}_1 + \boldsymbol{b}_1\right)\boldsymbol{W}_2 + \boldsymbol{b}_2 \tag{39}$$

$$= \boldsymbol{g}\left(\sum_{j=1}^{n} \boldsymbol{F}_i^{\mathrm{ATB}}(\boldsymbol{x}_j; \boldsymbol{X})\boldsymbol{W}_1 + \boldsymbol{b}^{\mathrm{ATB}}\boldsymbol{W}_1 + \boldsymbol{b}_1\right)\boldsymbol{W}_2 + \boldsymbol{b}_2 \tag{40}$$

$$= \boldsymbol{g}\left(\sum_{j=1}^{n} \boldsymbol{F}_i^{\mathrm{Pre}\,g}(\boldsymbol{x}_j) + \boldsymbol{b}^{\mathrm{Pre}\,g}\right)\boldsymbol{W}_2 + \boldsymbol{b}_2 \tag{41}$$

$$= \left(\sum_{j=1}^{n} \boldsymbol{h}_j\left(\boldsymbol{F}_i^{\mathrm{Pre}\,g}(\boldsymbol{x}_1), \ldots, \boldsymbol{F}_i^{\mathrm{Pre}\,g}(\boldsymbol{x}_n), \boldsymbol{b}^{\mathrm{Pre}\,g}; \widetilde{g}, \boldsymbol{0}\right)\right.$$

$$\left. + \boldsymbol{h}_b\left(\boldsymbol{F}_i^{\mathrm{Pre}\,g}(\boldsymbol{x}_1), \ldots, \boldsymbol{F}_i^{\mathrm{Pre}\,g}(\boldsymbol{x}_n), \boldsymbol{b}^{\mathrm{Pre}\,g}; \widetilde{g}, \boldsymbol{0}\right)\right)\boldsymbol{W}_2 + \boldsymbol{b}_2 \tag{42}$$

$$= \sum_{j=1}^{n} \boldsymbol{h}_j\left(\boldsymbol{F}_i^{\mathrm{Pre}\,g}(\boldsymbol{x}_1), \ldots, \boldsymbol{F}_i^{\mathrm{Pre}\,g}(\boldsymbol{x}_n), \boldsymbol{b}^{\mathrm{Pre}\,g}; \widetilde{g}, \boldsymbol{0}\right)\boldsymbol{W}_2 \tag{43}$$

$$+ \boldsymbol{h}_b\left(\boldsymbol{F}_i^{\mathrm{Pre}\,g}(\boldsymbol{x}_1), \ldots, \boldsymbol{F}_i^{\mathrm{Pre}\,g}(\boldsymbol{x}_n), \boldsymbol{b}^{\mathrm{Pre}\,g}; \widetilde{g}, \boldsymbol{0}\right)\boldsymbol{W}_2 + \boldsymbol{b}_2 \tag{44}$$

$$= \sum_j \boldsymbol{F}_i^{\mathrm{ATB+FF}}(\boldsymbol{x}_j; \boldsymbol{X}) + \boldsymbol{b}^{\mathrm{ATB+FF}} \tag{45}$$

$$\boldsymbol{F}_i^{\mathrm{Pre}\,g}(\boldsymbol{x}_j; \boldsymbol{X}) = \boldsymbol{F}_i^{\mathrm{ATB}}(\boldsymbol{x}_j; \boldsymbol{X})\boldsymbol{W}_1 \tag{46}$$

$$\boldsymbol{b}^{\mathrm{Pre}\,g} = \boldsymbol{b}^{\mathrm{ATB}}\boldsymbol{W}_1 + \boldsymbol{b}_1 \tag{47}$$

$$\boldsymbol{F}_i^{\mathrm{ATB+FF}}(\boldsymbol{x}_j; \boldsymbol{X}) = \boldsymbol{h}_j\left(\boldsymbol{F}_i^{\mathrm{Pre}\,g}(\boldsymbol{x}_1), \ldots, \boldsymbol{F}_i^{\mathrm{Pre}\,g}(\boldsymbol{x}_n), \boldsymbol{b}^{\mathrm{Pre}\,g}; \widetilde{g}, \boldsymbol{0}\right)\boldsymbol{W}_2 \tag{48}$$

$$\boldsymbol{b}^{\mathrm{ATB+FF}} = \boldsymbol{h}_b\left(\boldsymbol{F}_i^{\mathrm{Pre}\,g}(\boldsymbol{x}_1), \ldots, \boldsymbol{F}_i^{\mathrm{Pre}\,g}(\boldsymbol{x}_n), \boldsymbol{b}^{\mathrm{Pre}\,g}; \widetilde{g}, \boldsymbol{0}\right)\boldsymbol{W}_2 + \boldsymbol{b}_2. \tag{49}$$

Then, the ATBFF method quantifies how much the input $\boldsymbol{x}_j$ impacted the computation of the output $\boldsymbol{y}_i^{\mathrm{ATB+FF}}$ by the norm $\|\boldsymbol{F}_i^{\mathrm{ATB+FF}}(\boldsymbol{x}_j; \boldsymbol{X})\|$. Detailed decomposition of the activation function $\boldsymbol{g}$ is described in Appendix D.

## C.3 ATBFFRES

The residual connection (RES; Eq. 4) just performs the vector addition. So, ATB, FF, and RES2 also can be decomposed into a sum of transformed vectors collectively:

$$(\mathrm{RES2} \circ \mathrm{FF} \circ \mathrm{ATB})(\boldsymbol{x}_i; \boldsymbol{X}) = (\mathrm{FF} \circ \mathrm{ATB})(\boldsymbol{x}_i; \boldsymbol{X}) + \mathrm{ATB}(\boldsymbol{x}_i; \boldsymbol{X}) \tag{50}$$

$$= \sum_{j \neq i} \boldsymbol{F}_i^{\mathrm{ATB+FF}}(\boldsymbol{x}_j; \boldsymbol{X}) + \boldsymbol{F}_i^{\mathrm{ATB}}(\boldsymbol{x}_i; \boldsymbol{X}) + \boldsymbol{b}^{\mathrm{ATB+FF}} \tag{51}$$

$$= \sum_j \boldsymbol{F}_i^{\mathrm{ATB+FF+RES}}(\boldsymbol{x}_j; \boldsymbol{X}) + \boldsymbol{b}^{\mathrm{ATB+FF}} \tag{52}$$

$$\boldsymbol{F}_i^{\mathrm{ATB+FF+RES}}(\boldsymbol{x}_j; \boldsymbol{X}) = \begin{cases} \boldsymbol{F}_i^{\mathrm{ATB+FF}}(\boldsymbol{x}_j; \boldsymbol{X}) & (j \neq j) \\ \boldsymbol{F}_i^{\mathrm{ATB+FF}}(\boldsymbol{x}_i; \boldsymbol{X}) + F_i^{\mathrm{ATB}}(\boldsymbol{x}_i; \boldsymbol{X}) & (j = i) \end{cases}. \tag{53}$$

Then, the ATBFFRES method quantifies how much the input $\boldsymbol{x}_j$ impacted the computation of the output $\boldsymbol{y}_i^{\mathrm{ATB+FF+RES}}$ by the norm $\|\boldsymbol{F}_i^{\mathrm{ATB+FF+RES}}(\boldsymbol{x}_j; \boldsymbol{X})\|$.

## C.4 ATBFFRESLN

By exploiting the decomposition of LN (Eq. 31 and Eq. 32), entire Transformer layer (ATTN, RES1, LN1, FF, RES2, and LN2) can be decomposed into a sum of transformer vectors:

$$\text{Layer}(\boldsymbol{x}_i; \boldsymbol{X}) = \text{LN}\Big(\sum_j \boldsymbol{F}_i^{\text{ATB+FF+RES}}(\boldsymbol{x}_j; \boldsymbol{X}) + \boldsymbol{b}^{\text{ATB+FF}}\Big) \tag{54}$$

$$= \sum_j \boldsymbol{F}^{\text{LN}}\Big(\boldsymbol{F}_i^{\text{ATB+FF+RES}}(\boldsymbol{x}_j; \boldsymbol{X})\Big) + \boldsymbol{F}^{\text{LN}}\Big(\boldsymbol{b}^{\text{ATB+FF}}\Big) + \boldsymbol{\beta} \tag{55}$$

$$= \sum_j \boldsymbol{F}_i^{\text{Layer}}(\boldsymbol{x}_j; \boldsymbol{X}) + \boldsymbol{b}^{\text{Layer}} \tag{56}$$

$$\boldsymbol{F}_i^{\text{Layer}}(\boldsymbol{x}_j; \boldsymbol{X}) = \boldsymbol{F}^{\text{LN}}\Big(\boldsymbol{F}_i^{\text{ATB+FF+RES}}(\boldsymbol{x}_j; \boldsymbol{X})\Big) \tag{57}$$

$$\boldsymbol{b}^{\text{Layer}} = \boldsymbol{F}^{\text{LN}}\Big(\boldsymbol{b}^{\text{ATB+FF}}\Big) + \boldsymbol{\beta}. \tag{58}$$

Then, the ATBFFRESLN method quantifies how much the input $\boldsymbol{x}_j$ impacted the computation of the layer output $\boldsymbol{y}_i$ by the norm $\|\boldsymbol{F}_i^{\text{Layer}}(\boldsymbol{x}_j; \boldsymbol{X})\|$.

## D DETAILS OF DECOMPOSITION IN § 3

### D.1 FEATURE ATTRIBUTION METHODS

One of the major ways to interpret black-box deep learning models is measuring how much each input feature contributes to the output. Given a model $f \colon \mathbb{R}^n \to \mathbb{R}; \boldsymbol{x} \mapsto f(\boldsymbol{x})$ and a certain input $\boldsymbol{x}' = (x'_1, \ldots, x'_n)$, this approach decomposes the output $f(\boldsymbol{x}')$ into the sum of the contributions $c_i$ corresponding to each input feature $x'_i$:

$$f(\boldsymbol{x}') = c_1(\boldsymbol{x}'; f) + \cdots + c_n(\boldsymbol{x}'; f). \tag{59}$$

Interpretation methods with this approach are called feature attribution methods (Carvalho et al., 2019; Burkart & Huber, 2021), and typical methods include Integrated Gradients (Sundararajan et al., 2017) and Shapley value (Lloyd S., 1953). Comparison between Integrated Gradients and Shapley value are described in Appendix E.

### D.2 INTEGRATED GRADIENTS (IG)

IG is an excellent feature attribution method in that it has several desirable properties such as *completeness* (§ 3). Specifically, IG calculates the contribution of each input feature $x'_i$ by attributing the output at the input $\boldsymbol{x}' \in \mathbb{R}^n$ relative to a baseline input $\boldsymbol{b} \in \mathbb{R}^n$:

$$\text{IG}_i(\boldsymbol{x}'; f, \boldsymbol{b}) := (x'_i - b_i) \int_{\alpha=0}^1 \frac{\partial f}{\partial x_i}\Big|_{\boldsymbol{x} = \boldsymbol{b} + \alpha(\boldsymbol{x}' - \boldsymbol{b})} \, \mathrm{d}\alpha. \tag{60}$$

This contribution $\text{IG}_i(\boldsymbol{x}'; f, \boldsymbol{b})$ satisfies equation 59:

$$f(\boldsymbol{x}') = f(\boldsymbol{b}) + \sum_{j=1}^n \text{IG}_j(\boldsymbol{x}'; f, \boldsymbol{b}). \tag{61}$$

The first term $f(\boldsymbol{b})$ can be eliminated by selecting a baseline $\boldsymbol{b}$ for which $f(\boldsymbol{b}) = 0$ (see D.3).

### D.3 DECOMPOSITION OF GELU WITH IG

This paper aims to expand the norm-based analysis (Kobayashi et al., 2020), the interpretation method for Transformer, into the entire Transformer layer. However, a simple approach to decompose the network in closed form as in Kobayashi et al. (2020; 2021) cannot incorporate the activation function (GELU, Hendrycks & Gimpel, 2016) contained in FF[9]. This paper solves this problem by decomposing GELU with IG (Sundararajan et al., 2017).

---

[9]Many recent Transformers, including BERT and RoBERTa, employ GELU as their activation function.

GELU $\colon \mathbb{R} \to \mathbb{R}$ is defined as follows:

$$\text{GELU}(x) = \frac{x}{2}\left(1 + \frac{2}{\sqrt{\pi}}\int_0^{\frac{x}{\sqrt{2}}} e^{-t^2}\mathrm{d}t\right). \tag{62}$$

Considering that in the Transformer layer, the input $x \in \mathbb{R}$ of GELU can be decomposed into a sum of $x_j$ terms that rely on each token representation of layer inputs ($x = x_1 + \cdots + x_n$), GELU can be viewed as a multivariable function $\widetilde{\text{GELU}}$:

$$\widetilde{\text{GELU}}(x_1, \ldots, x_n) \coloneqq \text{GELU}(\textstyle\sum_{j=1}^n x_j). \tag{63}$$

Given a certain input $x' = x'_1 + \cdots + x'_n$, the contribution of each input feature $x'_j$ to the output $\text{GELU}(x')$ is calculated by decomposing $\widetilde{\text{GELU}}(x'_1, \ldots, x'_n)$ with IG (Eq. 60 and 61):

$$\begin{aligned}\text{GELU}(x') &= \widetilde{\text{GELU}}(x'_1, \ldots, x'_n) \\ &= \textstyle\sum_{j=1}^n \text{IG}_j(x'_1, \ldots, x'_n; \widetilde{\text{GELU}}, \mathbf{0}),\end{aligned} \tag{64}$$

where $\boldsymbol{b} = \mathbf{0}$ was chosen. Note that $\widetilde{\text{GELU}}(\boldsymbol{b}) = 0$ and the last term in equation 61 is eliminated.

**Decomposition of broadcasted GELU** In a practical neural network implementation, the GELU layer is passed a vector $\boldsymbol{x}_1 + \cdots + \boldsymbol{x}_n \in \mathbb{R}^{d'}$ instead of a scalar $x_1 + \cdots + x_n \in \mathbb{R}$ and the GELU function is applied (broadcasted) element-wise. Let $\mathbf{GELU} \colon \mathbb{R}^{d'} \to \mathbb{R}^{d'}$ be defined as the function that broadcasts the element-level activation $\text{GELU} \colon \mathbb{R} \to \mathbb{R}$. The contribution of each input vector $\boldsymbol{x}'_j = [x'_j[1], \ldots, x'_j[d']]$ to the output vector $\mathbf{GELU}(\boldsymbol{x}'_1 + \cdots + \boldsymbol{x}'_n)$ is as follows:

$$\mathbf{GELU}(\boldsymbol{x}'_1 + \cdots + \boldsymbol{x}'_n) = \begin{bmatrix} \text{GELU}(\boldsymbol{x}'_1[1] + \cdots + \boldsymbol{x}'_n[1]) \\ \vdots \\ \text{GELU}(\boldsymbol{x}'_1[d'] + \cdots + \boldsymbol{x}'_n[d']) \end{bmatrix}^\top \tag{65}$$

$$= \begin{bmatrix} \sum_{j=1}^n \text{IG}_j(\boldsymbol{x}'_1[1], \ldots, \boldsymbol{x}'_n[1]; \widetilde{\text{GELU}}, \mathbf{0}) \\ \vdots \\ \sum_{j=1}^n \text{IG}_j(\boldsymbol{x}'_1[d'], \ldots, \boldsymbol{x}'_n[d']; \widetilde{\text{GELU}}, \mathbf{0}) \end{bmatrix}^\top \tag{66}$$

$$= \sum_{j=1}^n \begin{bmatrix} \text{IG}_j(\boldsymbol{x}'_1[1], \ldots, \boldsymbol{x}'_n[1]; \widetilde{\text{GELU}}, \mathbf{0}) \\ \vdots \\ \text{IG}_j(\boldsymbol{x}_1[d'], \ldots, \boldsymbol{x}_n[d']; \widetilde{\text{GELU}}, \mathbf{0}) \end{bmatrix}^\top. \tag{67}$$

The above decomposition is applicable to any activation function $g$ that passes through the origin and is differentiable in practice, and covers all activation functions currently employed in Transformers, such as ReLU (Nair & Hinton, 2010), SiLU (Elfwing et al., 2018), and SwiGLU (Shazeer, 2020).

## E COMPARISON BETWEEN FEATURE ATTRIBUTION METHODS

We decomposed the nonlinear activation function using Integrated Gradients (IG), a feature attribution method (see § 3 and Appendix D), which is motivated by its desirable property *completeness* to be combined with norm-based analysis (§ 3). Nevertheless, there is another typical attribution method, Shapley Value (Lloyd S., 1953) (SV), that satisfies this property as well. In this section, we compare IG with SV in the context of our norm-based analysis.

### E.1 SHAPLEY VALUE

Shapley Value (SV) was originally introduced in cooperative game theory, and this concept has been imported into the machine learning field to compute specific input attribution. In our specific setting (Eq. 63), the sum of elements $x = x_1 + \ldots + x_n$ is fed into a nonlinear function (e.g., GELU). Each element $x_j[k]$ can be considered as a player in a cooperative game, and SV can

calculates its contribution to output. A well-known XAI method based on SV is SHapley Additive exPlanation (SHAP, Lundberg & Lee, 2017), which extends SV to apply to neural models. In our setting, SV and SHAP are equivalent (strictly speaking, SV is equivalent to SHAP with zero inputs as a reference input) because the target process can be viewed as a cooperative game as well as a neural model (since it starts with a sum of inputs).

### E.2 IMPLEMENTATION OF SV AND IG

**SV:** SV computation in Python can be achieved in several ways using the `shap` library (Lundberg & Lee, 2017)[10]. We used three SV methods: (i) Exact, (ii) Sampling, and (iii) Kernel methods. (i) Exact method is an exact computation of the Shapley value, computing the output of a nonlinear function for all combinations ($\mathcal{O}(2^n)$) of input elements ($x_1[k], ..., x_n[k]$) to compute the expected change in output with or without each element. (ii) Sampling method (Štrumbelj & Kononenko, 2010) approximates the computation with only the sampled combinations instead of all combinations. `shap` library samples $2 \times n + 2048$ combinations by default. (iii) Kernel method (Lundberg & Lee, 2017) formulates the SV calculation as a regression problem and approximates its calculation by sampling. It also samples $2 \times n + 2048$ combinations by default.

**IG:** Our IG computation was achieved in Python using `Captum` library (Kokhlikyan et al., 2020)[11]. In particular, the Gauss-Legendre quadrature method was used as the method of integral calculation (selectable argument).

### E.3 COMPARISON BETWEEN IG AND SV

We use IG and SV (three methods above) with our methods and compare the obtained performance and results. We experimented on NVIDIA GeForce RTX 2080 Ti and Intel Xeon Silver 4112.

First, we compared their speed (execution time) using BERT-tiny and three inputs of length $n = 5, 10, 15$. Any of the three SV methods required more time for longer input; specifically, these required more than 15 minutes to process the input of length $n = 15$, which is a typical (or shorter) length of a subword-segmented single sentence. On the other hand, IG required less than 1 second for any input of them (see Appendix F for detailed calculation cost). Hence, SV requires huge execution time for longer inputs and larger models, then SV could not be employed in this study.

Next, we compared their yielding results using BERT-tiny and 10 input of length $n \leq 15$ sampled from the validation set of SST-2 dataset. Fig. 11 shows results of macro contextualization change by FF, RES2, and LN2 (§ 5.1) for IG and SV. We can see that IG and SV (three methods) yielded similar results. This result suggests that the results and claims in this paper are not be significantly affected by the choice of attribution methods.

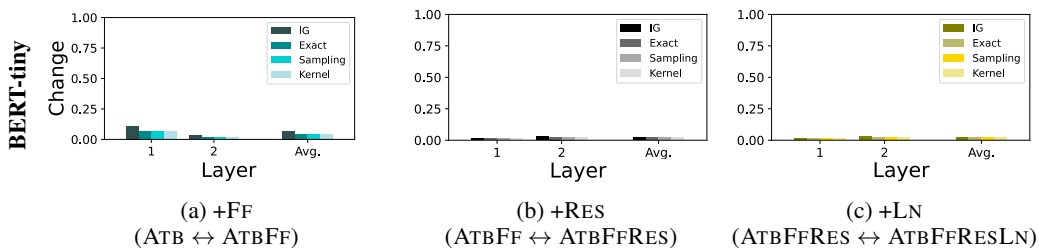

(a) +FF
(ATB ↔ ATBFF)

(b) +RES
(ATBFF ↔ ATBFFRES)

(c) +LN
(ATBFFRES ↔ ATBFFRESLN)

Figure 11: Contextualization changes between before and after each component in FFBs (FF, RES2, and LN2) of BERT-tiny calclulated with IG and SV (Exact, Sampling, and Kernel methods). The higher the bar, the more drastically the token-to-token contextualization (attention maps) changes due to the target component.

---

[10] https://github.com/shap/shap
[11] https://github.com/pytorch/captum

## F    COMPUTATIONAL COST OF OUR ANALYSIS

We test the computational costs of our method against inputs of various lengths in six BERT variants of different sizes. We fed each sample of the Wikipedia dataset into each BERT model and measured the time to run our analysis. We experimented on NVIDIA GeForce RTX 2080 Ti and Intel Xeon Silver 4112. Fig. 12 shows that the cost increase (running time) against input length becomes sharper when using larger models (hidden dimension and number of layers). That is, the cost increases interactively with the length of the input and the size of the model. This means that our method would be costly to apply to billion-scale LLMs, and lightening the computation is an important future work.

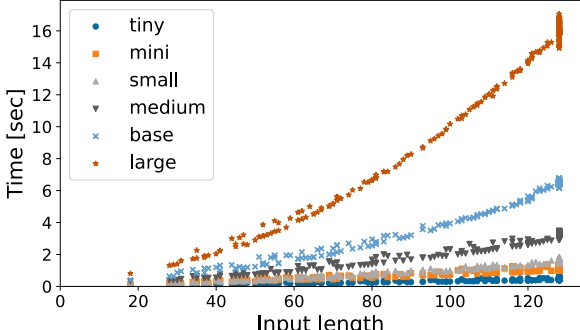

Figure 12: Running times of our analysis method for six BERT variants of different sizes and inputs of various lengths.

## G    SUPPLEMENTAL RESULTS OF CONTEXTUALIZATION CHANGE

We reported the contextualization change through each component in FFBs of BERT-base and GPT-2 on the Wikipedia dataset in § 5. We will report the results of the other models/datasets in this section. In addition, we provide full results of linguistic patterns in FF's contextualization effects that could not be included in the main body.

### G.1    MACRO CONTEXTUALIZATION CHANGE

The contextualization changes through each component in FFBs of six variants of BERT models with different sizes are shown in Fig. 14. The results for four variants of BERT-base models trained with different seeds are shown in Fig. 15. The results for two variants of RoBERTa models with different sizes are shown in Fig. 16. The results for OPT 125M model are shown in Fig. 17. The results for BERT-base and GPT-2 on the SST-2 dataset are shown in Fig. 18. These different settings with other models/datasets also yield similar results reported in § 5.1: each component did modify the input contextualization, especially in particular layers. The mask language models showed a consistent trend of larger changes by FF and LN in the middle to late layers. On the other hand, the causal language models showed a trend of larger changes by FF in the early layers.

### G.2    LINGUISTIC PATTERNS IN FF'S CONTEXTUALIZATION EFFECTS

Tables 2 and 3 are the extended versions of Table 1 showing the top ten word-word pairs with the highest FF-amp scores in each layer of GPT-2 and BERT.

Fig. 13 shows the distribution of word-word pair types (e.g., composing the same word) in the top 50 token pairs with the highest FF-amp score, which is measured on the SST-2 dataset. While the overall trend is similar to the results on Wikipedia dataset (Fig. 5), there are some minor differences (e.g., subword and same token categories). These differences can be attributed to differences in the datasets. The Wikipedia dataset is composed of excerpts from Wikipedia articles, while the SST-2 dataset is composed of movie reviews. Experimenting with other datasets and models is future work.

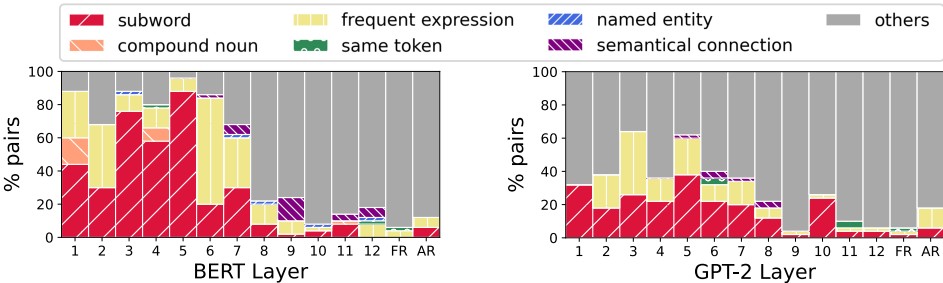

Figure 13: Breakdown of the category labels we manually assigned to top 50 pairs having the largest FF-amp score in each layer of BERT and GPT-2 with SST-2 dataset. We also assigned the labels to fully random 50 pairs ("FR") and adjacent random 50 pairs ("AR").

## H    SUPPLEMENTAL RESULTS OF DYNAMICS OF CONTEXTUALIZATION CHANGE

We reported the dynamics of contextualization change of BERT-base and GPT-2 on the Wikipedia dataset in § 6. We will report the results of the other models/datasets in this section.

### H.1    CONTEXTUALIZATION CHANGE THROUGH FF AND SUBSEQUENT COMPONENTS

The contextualization changes through by FF and subsequent components, RES and LN, in six variants of BERT models with different sizes are shown in Fig. 19. The results for four variants of BERT-base models trained with different seeds are shown in Fig. 20. The results for two variants of RoBERTa models with different sizes are shown in Fig. 21. The results for OPT 125M model are shown in Fig. 22. The results for BERT-base and GPT-2 on the SST-2 dataset are shown in Fig. 23. These different settings with other models/datasets also yield similar results reported in § 6: through the FF and subsequent RES and LN, the contextualization change score once becomes large but finally converges to be small; that is, the contextualization by FFs tends to be canceled by the following components.

### H.2    FF AND RES

The L2 norm of the output vectors from FF and the vectors bypassed via RES2, in six variants of BERT models with different sizes are shown in Fig. 24. The results for four variants of BERT-base models trained with different seeds are shown in Fig. 25. The results for two variants of RoBERTa models with different sizes are shown in Fig. 26. The results for BERT-base and GPT-2 on the SST-2 dataset are shown in Fig. 27. These different settings with other models/datasets also yield similar results reported in § 6.1: the vectors bypassed via RES2 are more than twice as large as output vectors from FF in the L2 norm in more than half of the layers. That is, the representation (contextualization) updated by the FF tends to be overwritten/canceled by the original one.

### H.3    FF AND LN

Mean absolute value in each dimension of the input/output vectors of FF and the weight parameter $\gamma$ of LN at the layer where FF's contextualization effects are strongly cancelled by LN, in six variants of BERT models with different sizes are shown in Fig. 28. The results for four variants of BERT-base models trained with different seeds are shown in Fig. 29. The results for two variants of RoBERTa models with different sizes are shown in Fig. 30. The results for BERT-base and GPT-2 on the SST-2 dataset are shown in Fig. 31. These different settings with other models/datasets also yield similar results reported in § 6.2: the FF output representation has outliers in some specific dimensions (green lines in the figures), and the weight $\gamma$ of LN tends to shrink these special dimensions (red lines in the figures). In the layers where FF incurs a relatively large impact on contextualization, the Pearson correlation coefficient between LN's $\gamma$ and mean absolute value of FF output by dimension was from $-0.38$ to $-0.93$ in BERT-large, from $-0.32$ to $-0.47$ in BERT-medium, from $-0.56$ to $-0.76$ in BERT-small, from $-0.55$ to $-0.73$ in BERT-mini, $-0.71$ in BERT-tiny, from $-0.69$ to $-0.74$

in BERT-base (seed 0), from $-0.48$ to $-0.74$ in BERT-base (seed 10), from $-0.50$ to $-0.68$ in BERT-base (seed 20). In these layers of RoBERTa models, the Pearson's $r$ was small: from $-0.02$ to $-0.28$ in RoBERTa-large and from $0.01$ to $-0.14$ in RoBERT-base. However, the Spearman's $\rho$ was large: from $-0.46$ to $-0.56$ in RoBERTa-large and from $0.80$ to $-0.94$ in RoBERT-base.

In § 6.2, we also observed that ignoring such special outlier dimensions (bottom 1% with the lowest value of $\gamma$) in calculating FF's contextualization makes the change score quite small. The contextualization changes by FF when ignoring the dimensions are shown in Fig. 32.

Table 2: Word pairs for which FF amplified the interaction the most in BERT. The text colors are aligned with word pair categories: subword, compound noun, common expression, same token, named entity, and others.

| Layer | Top amplified token-pairs |
|---|---|
| 1 | (##our, det), (##iques, ##mun), (##cend, trans), (outer, space), (##ili, res), (##ific, honor), (##nate, ##imi), (opera, soap), (deco, art), (##night, week) |
| 2 | (##roy, con), (guard, national), (america, latin), (easily, could), (##oy, f), (marshall, islands), (##ite, rec), (channel, english), (finance, finance), (##ert, rev) |
| 3 | (', t), (##mel, ##ons), (hut, ##chin), (water, ##man), (paso, ##k), (avoid, ##ant), (toys, ##hop), (competitive, ##ness), (##la, ##p), (##tree, ##t) |
| 4 | (##l, ds), (##l, ##tera), (##cz, ##ave), (##r, ##lea), (##e, beth), (##ent, ##ili), (##er, ##burn), (##et, mn), (##ve, wai), (##rana, ke) |
| 5 | (##ent, ##ili), (##ious, ##car), (##on, ##ath), (##l, ds), (##r, ##ense), (##ence, ##ili), (res, ##ili), (##rative, ##jo), (##ci, ##pres), (##able, ##vor) |
| 6 | (on, clay), (with, charged), (##ci, ##pres), (be, considers), (on, behalf), (fleming, colin), (##r, ##lea), (-, clock), (##ur, nam), (by, denoted) |
| 7 | (##ons, ##mel), (vi, saint), (vi, st), (##l, ##ife), (##ti, jon), (##i, wii), (##en, ##chel), (##son, bis), (vessels, among), (##her, ##rn) |
| 8 | (##ano, ##lz), (bo, ##lz), (nathan, or), (arabia, against), (sant, ##ini), (previous, unlike), (saudi, against), (##ia, ##uring), (he, gave), (tnt, equivalent) |
| 9 | (no, situation), (##ek, czech), (according, situation), (decided, year), (v, classification), (eventually, year), (but, difficulty), (##ference, track), (she, teacher), (might, concerned) |
| 10 | (jan, ##nen), (hee, ## ], (primary, stellar), (crete, crete), (nuclear, 1991), (inspector, police), (##tu, ##nen), (quote, quote), (f, stellar), (v, stellar) |
| 11 | (tiny, tiny), (##water, ##water), (hem, hem), (suddenly, suddenly), (fine, singer), (henley, henley), (highway, highway), (moving, moving), (dug, dug), (farmers, agricultural) |
| 12 | (luis, luis), (tong, 同), (board, judicial), (##i, index), (一, 一), (four, fifty), (cloud, thunder), (located, transmitter), (##ota, ##ɔ), (##ss, analysis) |

Table 3: Word pairs for which FF amplified the interaction the most in GPT-2. The text colors are aligned with word pair categories: subword, compound noun, common expression, same token, named entity, and others.

| Layer | Top amplified token-pairs |
|---|---|
| 1 | (ies, stud), (ning, begin), (ever, how), (ents, stud), (ited, rec), (une, j), (al, sever), (itions, cond), (ang, p), (ree, c) |
| 2 | (al, sever), (itions, cond), (z, jan), (ning, begin), (_was, this), (er, care), (ies, stud), (une, j), (s, 1990), (ents, stud) |
| 3 | (_than, _rather), (z, jan), (_then, since), (bin, ro), (_long, _how), (ting, _rever), (une, j), (_others, _among), (arks, _cl), (_into, _turning) |
| 4 | (_same, _multiple), (ouri, _miss), (_jung, lex), (_b, _strength), (_others, _among), (_based, _".), (adesh, hra), (_top, _among), (t, _give), (bin, ro) |
| 5 | (ol, _brist), (ol, ink), (gal, _man), (ac, _pens), (op, _link), (_whit, _&), (ant, _avoid), (_people, _important), (un, _bra), (_1995, _1994) |
| 6 | (ol, ink), (_del, del), (o, _dec), (ist, oh), (it, _me), (_route, route), (thel, in), (ord, m), (adier, division), (we, ob) |
| 7 | (_ve, _las), (che, _ro), (43, _âĢ), (_green, _mark), (_cer, _family), (_jack, eter), (orses, '), (_kilometres, _lies), (,, _disappointed), (umer, in) |
| 8 | (_ar, _goddess), (_d, ight), (_33, _density), (_cer, _family), (_08, _2007), (bid, _family), (_14, _density), (_e, _family), (_50, _yield), (_15, _density) |
| 9 | (_ray, _french), (_11, uly), (_one, _score), (on, _french), (_41, _%), (_37, _%), (_equipped, _another), (bc, _;), (ad, en), (.,, te) |
| 10 | (ho, ind), (_by, ashi), (_bra, _von), (_and, _317), (ad, en), (_loves, ashi), (_u, asaki), (_he, ner), (_land, ines), (_de, die) |
| 11 | (it, '), (uman, _when), (ave, _daughters), (_bird, _:), (ic, there), (ia, _counties), (bc, _;), (_and, had), (_2010, _us), (ĩ, :) |
| 12 | (_operational, _not), (_answered, _therefore), (_degrees, _having), (_varying, _having), (k, rant), (_supplied, _also), (_daring, _has), (_prominent, 's), (a, ually), (_stress, _:) |

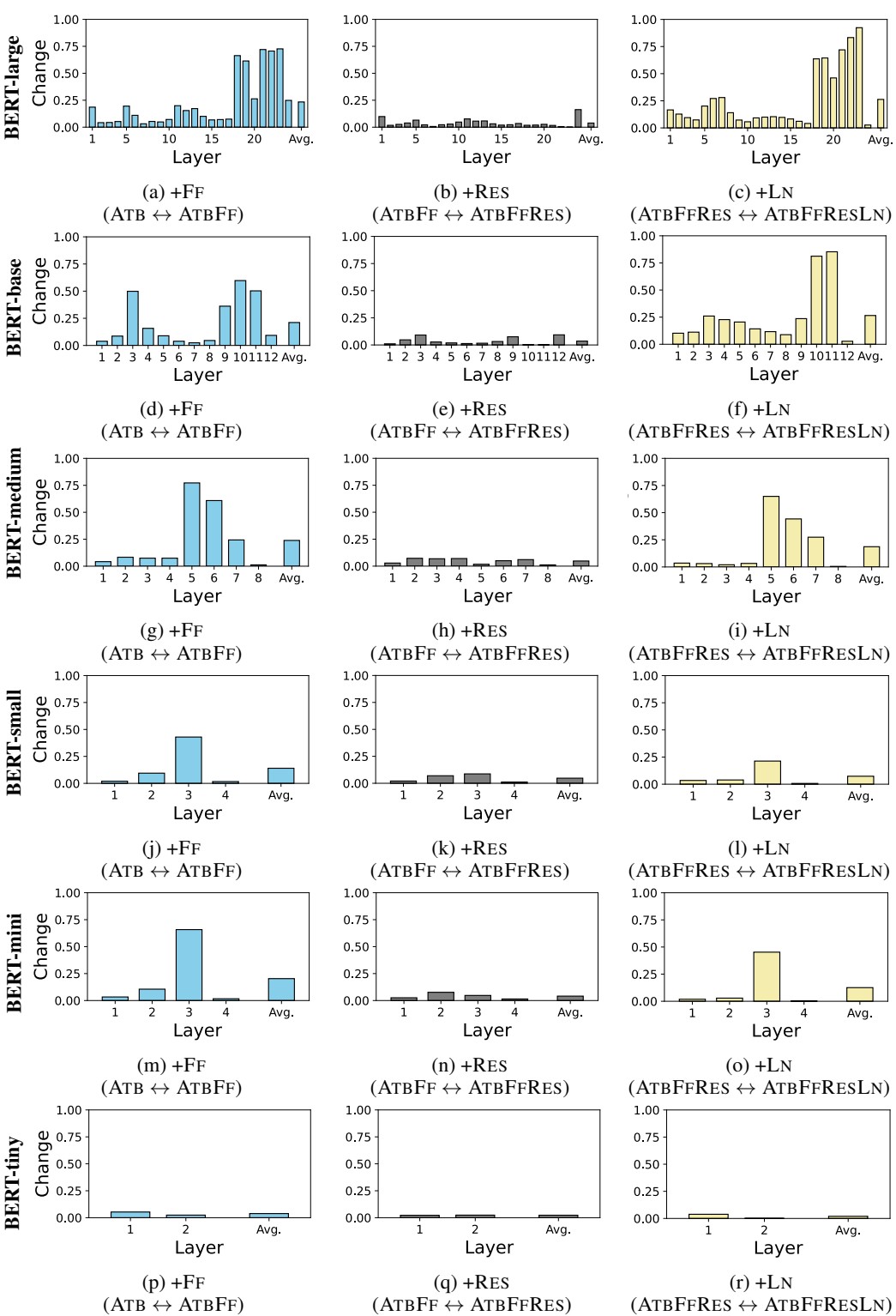

Figure 14: Contextualization changes between before and after each component in FFBs (FF, RES2, and LN2) of six variants of BERT models with different sizes (large, base, medium, small, mini, and tiny). The higher the bar, the more drastically the token-to-token contextualization (attention maps) changes due to the target component.

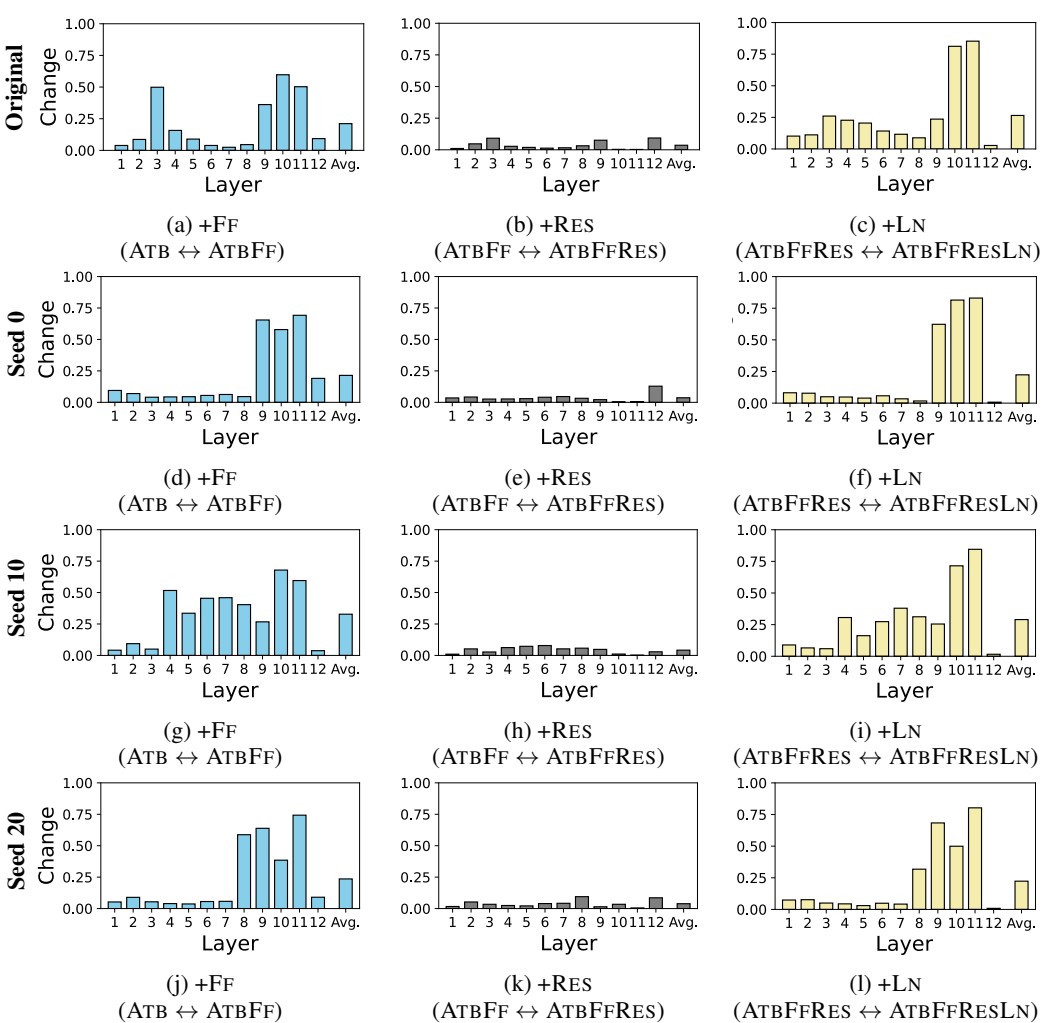

Figure 15: Contextualization changes between before and after each component in FFBs (FF, RES2, and LN2) of four variants of BERT-base models trained with different seeds (original, 0, 10, and 20). The higher the bar, the more drastically the token-to-token contextualization (attention maps) changes due to the target component.

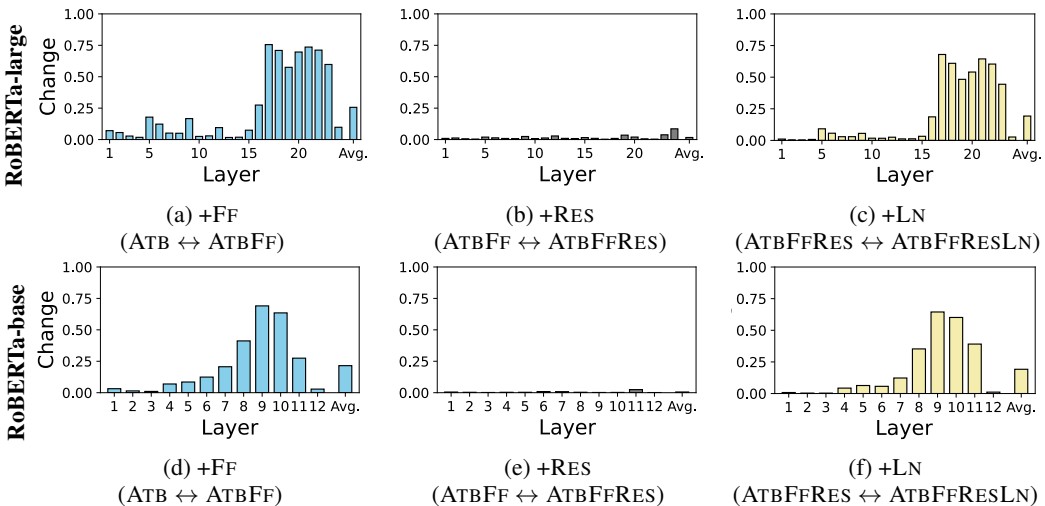

Figure 16: Contextualization changes between before and after each component in FFBs (FF, RES2, and LN2) of two variants of RoBERTa models with different sizes (large and base). The higher the bar, the more drastically the token-to-token contextualization (attention maps) changes due to the target component.

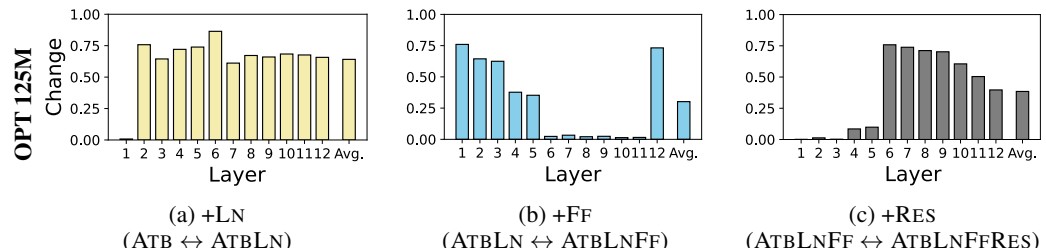

Figure 17: Contextualization changes between before and after each component in FFBs (FF, RES2, and LN2) of OPT 125M model. The higher the bar, the more drastically the token-to-token contextualization (attention maps) changes due to the target component.

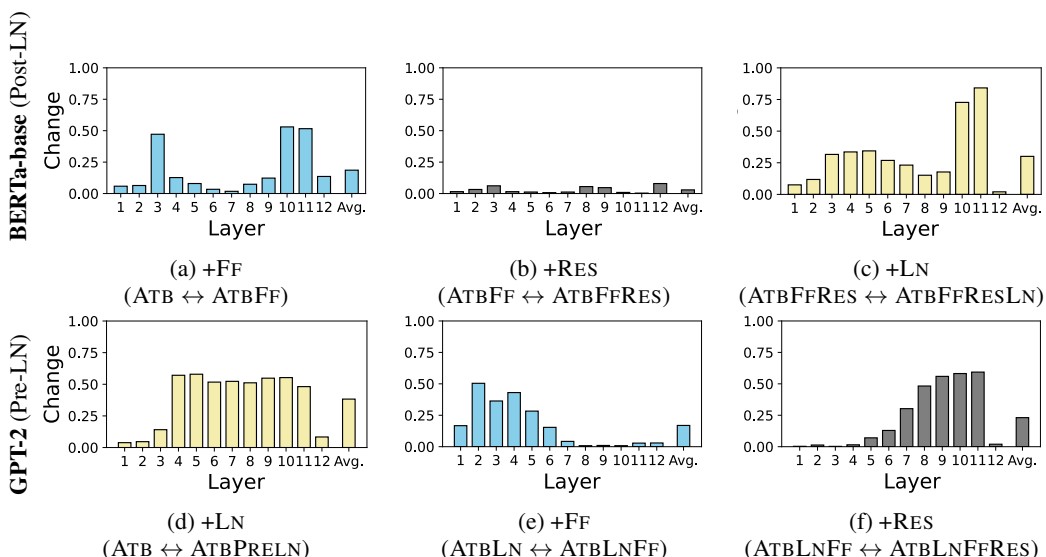

Figure 18: Contextualization changes between before and after each component in FFBs (FF, RES2, and LN2) of BERT-base and GPT-2 on SST-2 dataset. The higher the bar, the more drastically the token-to-token contextualization (attention maps) changes due to the target component.

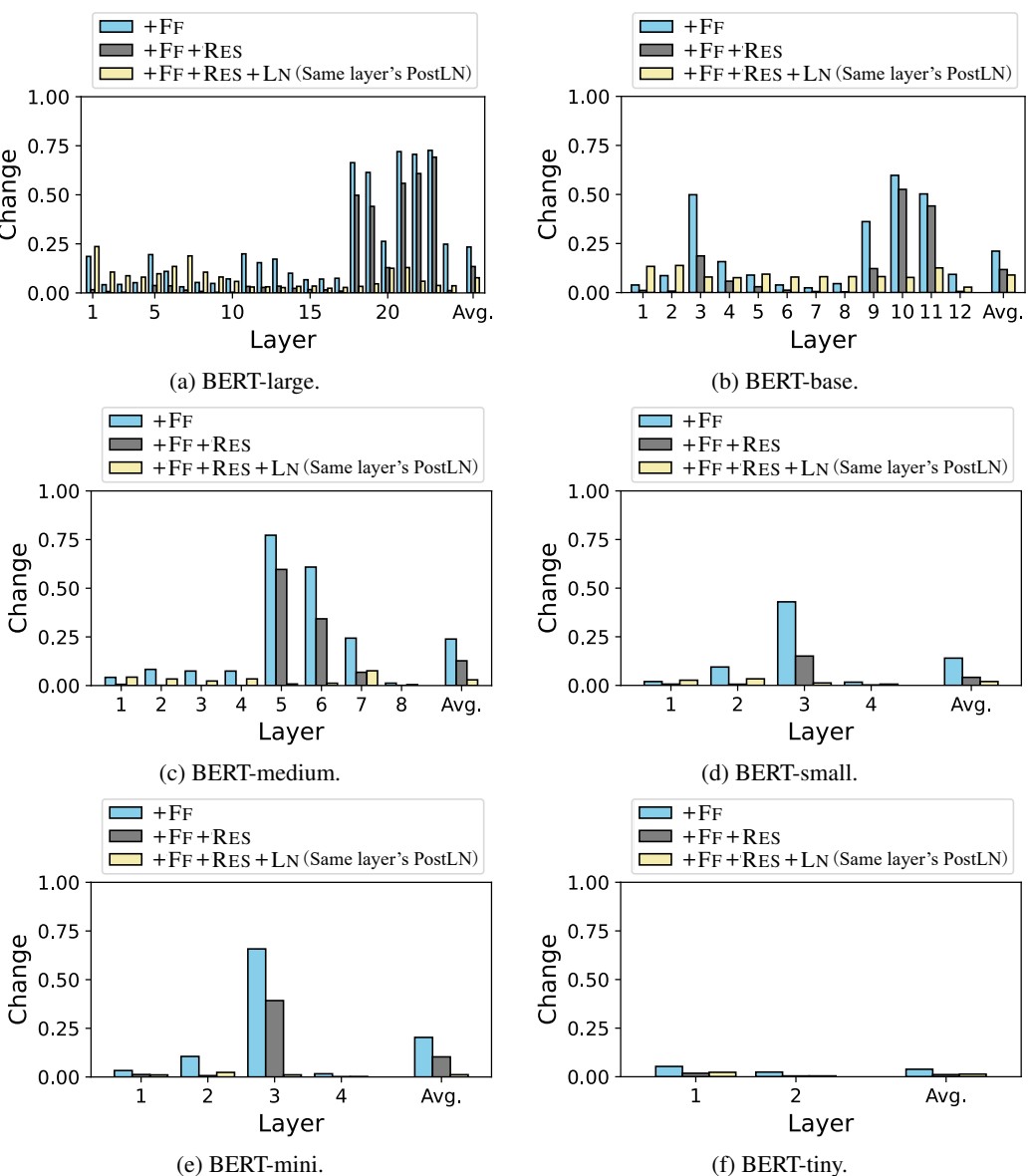

Figure 19: Contextualization changes through processing each component (FF, RES, and LN) from just before FF (ATB) in six variants of BERT models with different sizes (large, base, medium, small, mini, and tiny). The higher the bar, the more the contextualization (attention map) changes from just before FF.

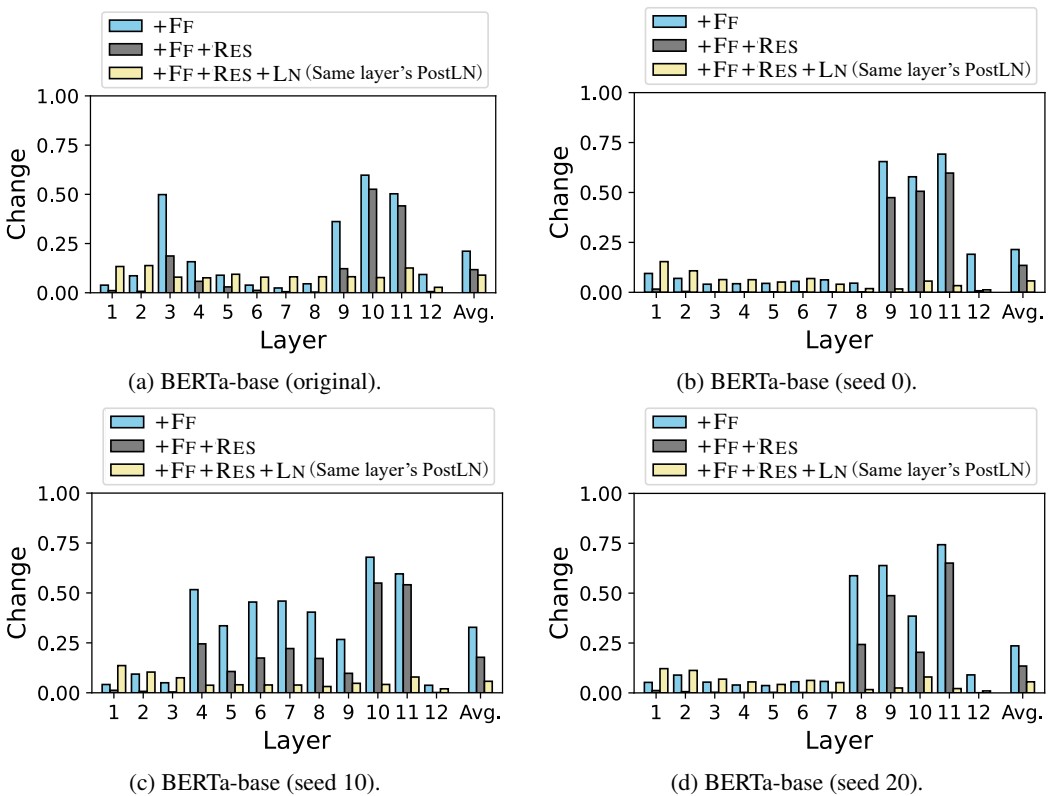

Figure 20: Contextualization changes through processing each component (FF, RES, and LN) from just before FF (ATB) in four variants of BERT-base models trained with different seeds (original, 0, 10, and 20). The higher the bar, the more the contextualization (attention map) changes from just before FF.

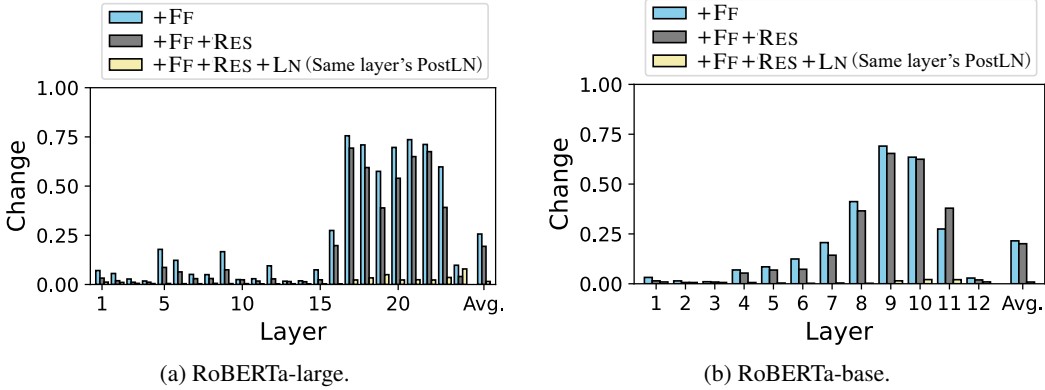

Figure 21: Contextualization changes through processing each component (FF, RES, and LN) from just before FF (ATB) in two variants of RoBERTa models with different sizes (large and base). The higher the bar, the more the contextualization (attention map) changes from just before FF.

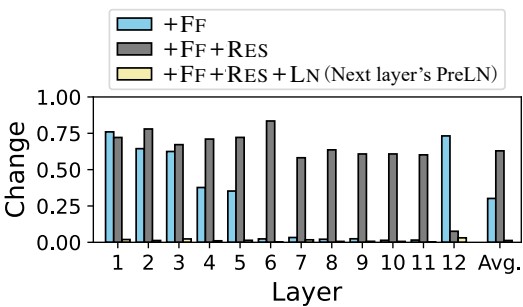

Figure 22: Contextualization changes through processing each component (FF, RES, and LN) from just before FF (ATBLN) in OPT 125M model. The higher the bar, the more the contextualization (attention map) changes from just before FF.

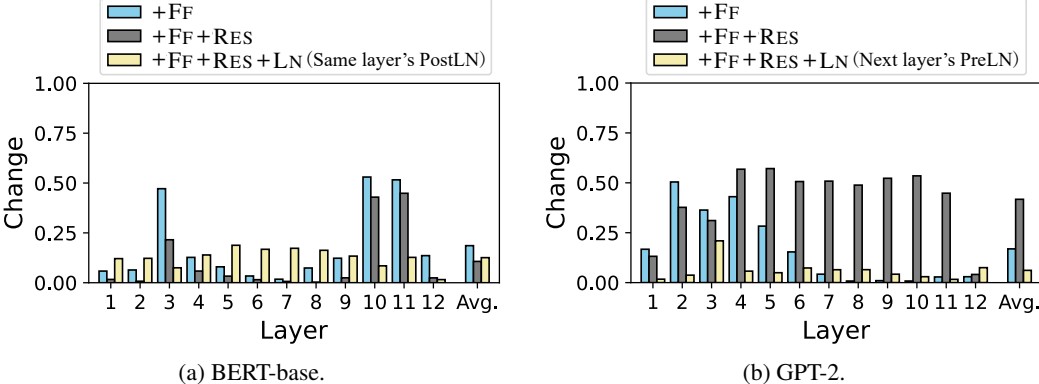

(a) BERT-base.

(b) GPT-2.

Figure 23: Contextualization changes through processing each component (FF, RES, and LN) from just before FF (ATB) in BERT-base and GPT-2 on SST-2 dataset. The higher the bar, the more the contextualization (attention map) changes from just before FF.

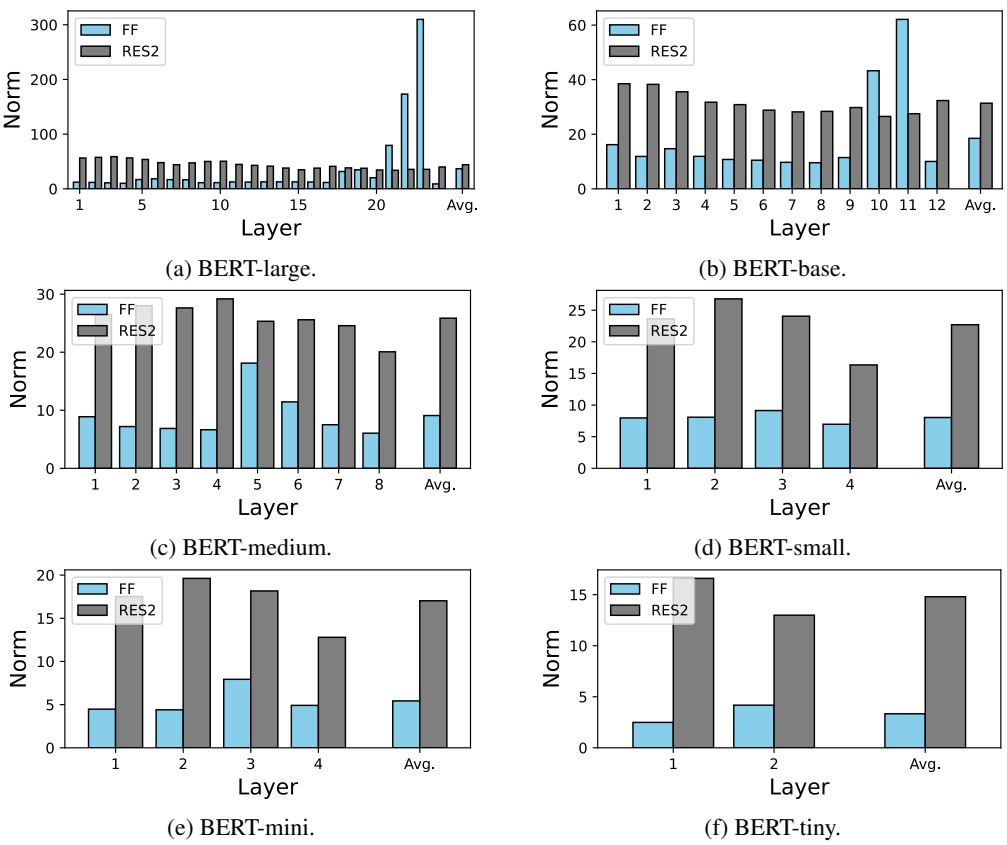

Figure 24: Averaged norm of the output vectors from FF and the bypassed vectors via RES2, calculated on the Wikipedia data for six variants of BERT models with different sizes (large, base, medium, small, mini, and tiny).

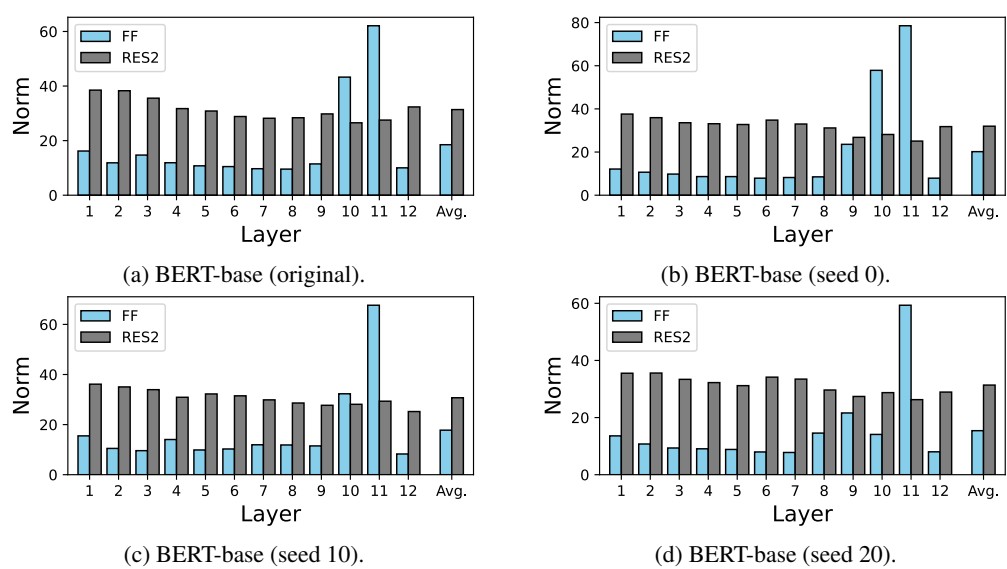

Figure 25: Averaged norm of the output vectors from FF and the bypassed vectors via RES2, calculated on the Wikipedia data for four variants of BERT-base models trained with different seeds (original, 0, 10, and 20).

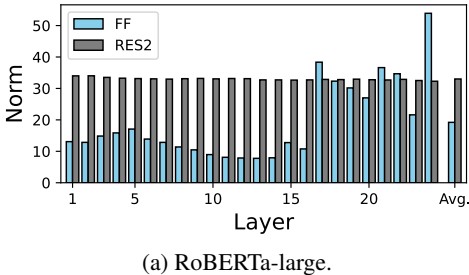

(a) RoBERTa-large.

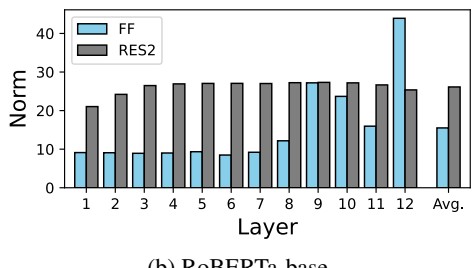

(b) RoBERTa-base.

Figure 26: Averaged norm of the output vectors from FF and the bypassed vectors via RES2, calculated on the Wikipedia data for two variants of RoBERTa models with different sizes (large and base).

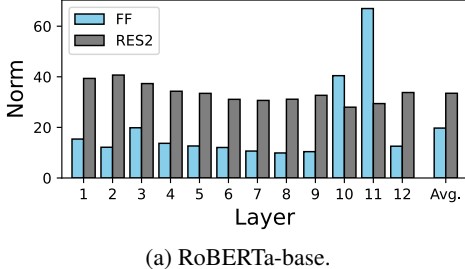

(a) RoBERTa-base.

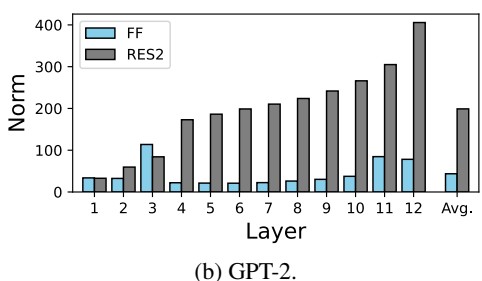

(b) GPT-2.

Figure 27: Averaged norm of the output vectors from FF and the bypassed vectors via RES2, calculated on the SST-2 dataset for BERT-base and GPT-2.

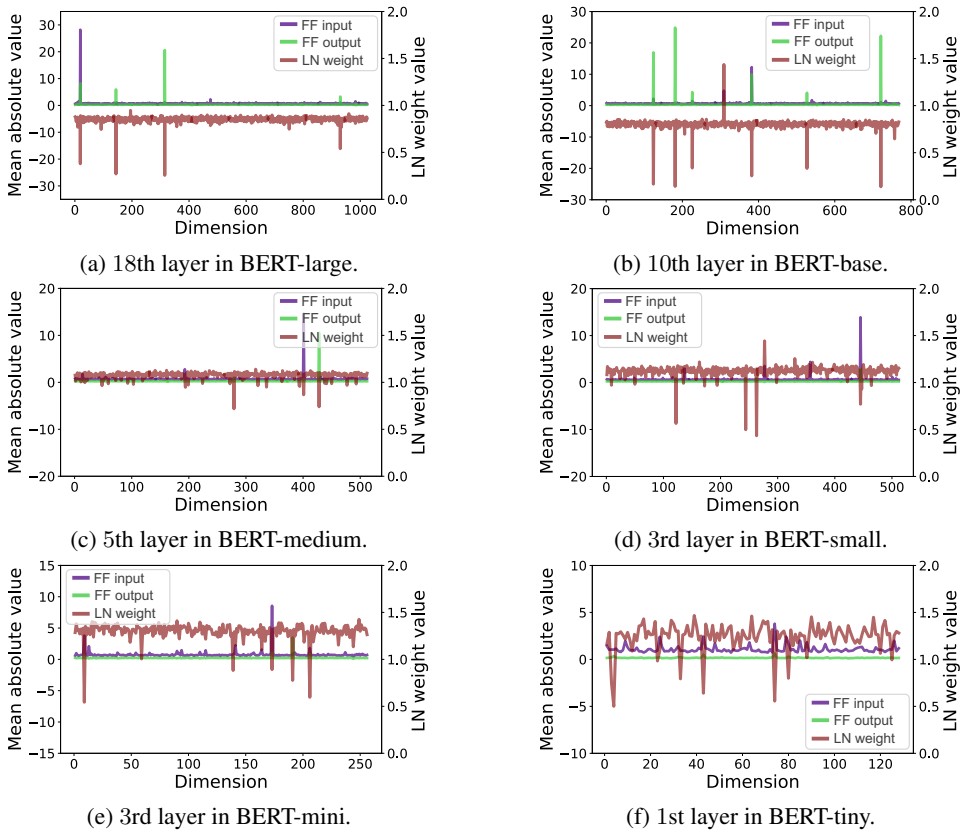

(a) 18th layer in BERT-large.

(b) 10th layer in BERT-base.

(c) 5th layer in BERT-medium.

(d) 3rd layer in BERT-small.

(e) 3rd layer in BERT-mini.

(f) 1st layer in BERT-tiny.

Figure 28: Mean absolute value in each dimension of the input/output vectors of FF across the Wikipedia data and the LN weight values at the certain layer of six variants of BERT models with different sizes (large, base, medium, small, mini, and tiny).

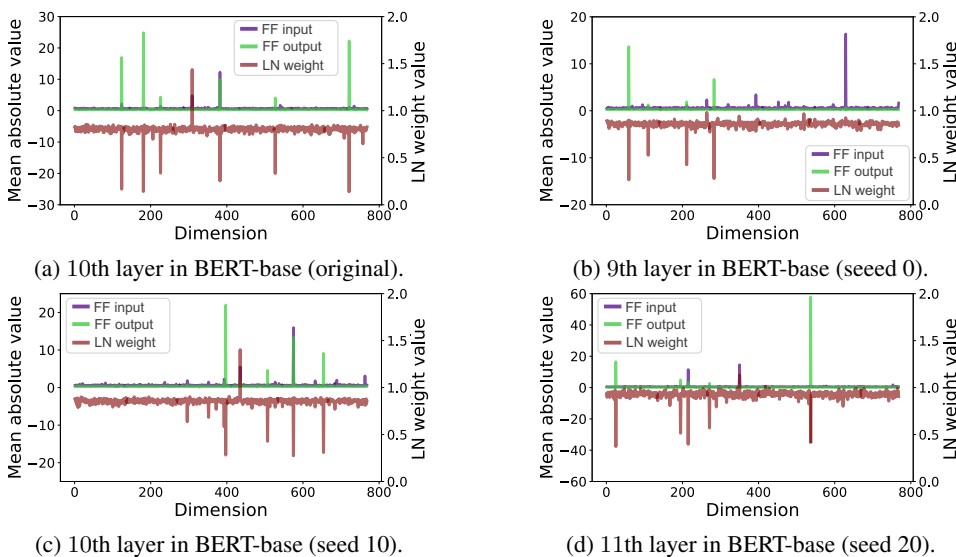

(a) 10th layer in BERT-base (original).

(b) 9th layer in BERT-base (seeed 0).

(c) 10th layer in BERT-base (seed 10).

(d) 11th layer in BERT-base (seed 20).

Figure 29: Mean absolute value in each dimension of the input/output vectors of FF across the Wikipedia data and the LN weight values at the certain layer of four variants of BERT-base models trained with different seeds (original, 0, 10, and 20).

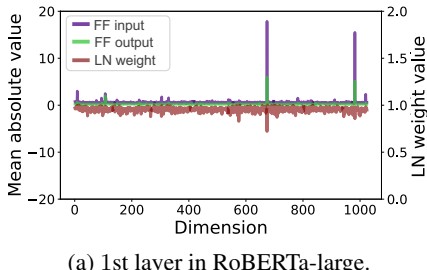

(a) 1st layer in RoBERTa-large.

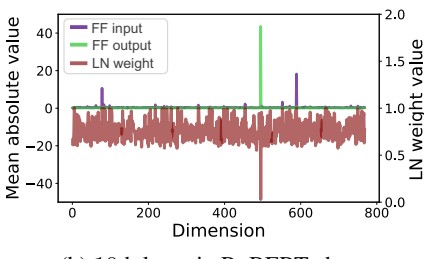

(b) 10th layer in RoBERTa-base.

Figure 30: Mean absolute value in each dimension of the input/output vectors of FF across the Wikipedia data and the LN weight values at the certain layer of two variants of RoBERTa with different sizes (large and base).

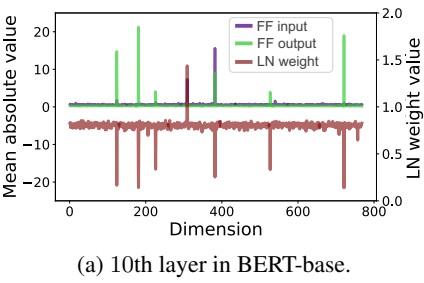

(a) 10th layer in BERT-base.

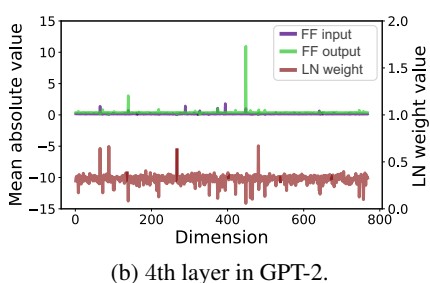

(b) 4th layer in GPT-2.

Figure 31: Mean absolute value in each dimension of the input/output vectors of FF across the SST-2 dataset and the LN weight values at the certain layer.

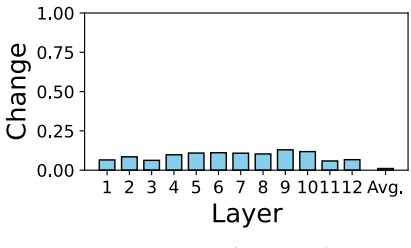

(a) ATB ↔ ATBFF in BERT-base.

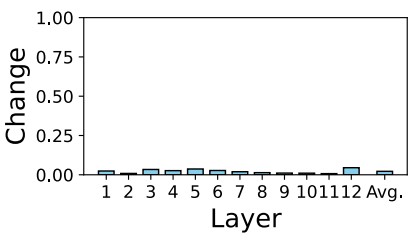

(b) ATBFF ↔ ATBFFRES in GPT-2

Figure 32: Contextualization changes by FF of BERT-base and GPT-2 on Wikipedia dataset when the 1% (seven) dimensions with the smallest LN weights $\gamma$ values is ignored in the norm calculation. The higher the bar, the more drastically the token-to-token contextualization (attention maps) changes due to the target component.

