# OpenReview forum: "Analyzing Feed-Forward Blocks in Transformers through the Lens of Attention Maps"
_ICLR.cc/2024/Conference — ICLR 2024 spotlight_

### Official Review · Reviewer_GB37 · 2023-10-26

**Soundness:** 3 good
**Presentation:** 3 good
**Contribution:** 3 good
**Rating:** 6
**Confidence:** 3

**Summary:**

This paper focuses on analyzing Feed-Forward (FF) blocks in the Transformer model, specifically regarding their impact on input contextualization. The authors utilize a refined attention map by combining norm-based analysis and the integrated gradient method, which offers completeness in understanding the FF block's behavior. In the experiment, Wikipedia excerpts and Stanford Sentiment Treebank v2 datasets are used, and the authors analyzed 11 11 masked LMs and one casual LM. The results show that FF blocks do modify input contextualization by amplifying specific linguistic compositions, such as subword pairs forming a single word. Furthermore, the authors discover that the FF block and its surrounding components tend to cancel out each other's contextualization effects, shedding light on the mechanism and suggesting redundancy in processing within the Transformer layer.

**Strengths:**

1. Analyzing Feed-Forward (FF) blocks in Transformer models through a refined attention map is novel, which combines norm-based analysis and the integrated gradient method. While previous research has explored the behavior of FF blocks, this study offers a unique perspective by leveraging the aforementioned techniques to gain a comprehensive understanding of their impact on input contextualization.

2. The experiments conducted with masked and causal language models demonstrate the effectiveness of the approach in capturing the modification of input contextualization by FF blocks. The clarity and precision of the analysis enhance the quality of the research, ensuring reliable and valid results.

3. The paper is clear and easy to follow. The description of the refined attention map, norm-based analysis, and integrated gradient method is presented in a clear and understandable manner. The experiments and their results are well-explained, enabling readers to grasp the implications of FF block behavior on input contextualization.

**Weaknesses:**

1. To enhance the clarity of the proposed method, it would be beneficial to provide a running example that illustrates the step-by-step process. By walking readers through a concrete example, they can more easily grasp the methodology and its application.

2. To better ground the paper in the existing literature, it would be valuable to provide a more detailed and comprehensive literature review. By thoroughly reviewing relevant prior research, the paper can establish its position within the broader academic discourse and highlight its unique contributions.

**Questions:**

What could be the challenges of applying the proposed method in larger models, such as OPT, LLaMA, as mentioned in the future work?

---

> ### Author Response · Authors · 2023-11-20
> **Response to Official Review by Reviewer GB37**
>
> We appreciate your constructive comments!
>
> # Providing running examples
> > Weaknesses 1
> > To enhance the clarity of the proposed method, it would be beneficial to provide a running example that illustrates the step-by-step process. By walking readers through a concrete example, they can more easily grasp the methodology and its application.
>
> We agree that providing concrete examples would be helpful to readers. We will add them to the camera-ready version.
>
> # More comprehensive review of existing literature
> > Weaknesses 2
> > To better ground the paper in the existing literature, it would be valuable to provide a more detailed and comprehensive literature review. By thoroughly reviewing relevant prior research, the paper can establish its position within the broader academic discourse and highlight its unique contributions.
>
> As for the additional literature related to this study, for example, there is research on not only reverse engineering of transformer models (mechanistic interpretability) [Elhage+’21,’22] but also research on comparing transformer models with human behavior [Oh&Schuler’22]. Unfortunately, due to space limitations, it is not immediately possible to add them to the paper; we will try to enhance the Introduction and Related work to be more comprehensive as much as possible in the camera-ready version.
>
> [Elhage+’21] A Mathematical Framework for Transformer Circuits (Anthropic) https://transformer-circuits.pub/2021/framework/index.html
> [Elhage+’22] Softmax Linear Units (Anthropic) https://transformer-circuits.pub/2022/solu/index.html
> [Oh&Schuler’22] Entropy- and Distance-Based Predictors From GPT-2 Attention Patterns Predict Reading Times Over and Above GPT-2 Surprisal (EMNLP 2022) https://aclanthology.org/2022.emnlp-main.632/
>
> # Challenges in applying to larger models
> > Questions
> > What could be the challenges of applying the proposed method in larger models, such as OPT, LLaMA, as mentioned in the future work?
>
> Although we used relatively small transformer models to demonstrate our proposed analysis method, analyzing LLMs is an important future work. Our method can be extended to LLMs without any technical modification, but it generally requires a computational cost typically beyond an academic resource. Formally, given an input of length $N$ and a model containing FF's intermediate dimension $d$, estimating one attention map requires $Nd$ times Integrated Gradients computation.

---

### Official Review · Reviewer_wwew · 2023-11-09

**Soundness:** 3 good
**Presentation:** 3 good
**Contribution:** 2 fair
**Rating:** 6
**Confidence:** 4

**Summary:**

This paper proposes to use attention map to analyze the effects of feed-forward blocks in transformers. Different from previous works which mostly focus on studying attention weights, the authors leverage norm-based analysis and integrated gradient methods on the the effect of feed-forward networks, residual connection, and layer normalization. Experiments done one different models (including different sizes of BERT and RoBERTa) and on different dataset (Wikipedia and Stanford Sentiment Treebank) suggest that feedforward networks (FF) amplify specific types of linguistic compositions, and its surrounding components tend to cancel out each other's contextualization effects.

**Strengths:**

1. This paper studies the contextualization effects of feed-forward blocks by leveraging attention maps, which were mostly ignored before (by only looking at attention weights). This presents new views on analyzing what each block in transformer is functioning.
2. The findings that FF and surrounding components tend to cancel out each other's effects are interesting. This may provide more perspectives in designing and training transformer models.

**Weaknesses:**

1. The paper suggests that because of the cancelled out effects in surrounding components there is "potential redundancy in transformer layers". However, there is not enough evidence to justify this claim (e.g., by training a transformer model removing some of the components). Without more experiments and results, it is not convincing what the conclusion of this paper is. More importantly, there is no systematic evaluation on the linguistic patterns (e.g., linguistic patterns distribution from different datasets on each layer) apart from some sampled amplified pairs. The results presented in Table 1 and Figure 5 are not evident.
2. The results on different BERT sizes, and different seeds do not seem to be always consistent (e.g., Figure 9, 10 in the appendix).

**Questions:**

1. Why is b set to a zero vector in equation 12?
2. Why do you think the patterns of changes between BERT and GPT-2 are quick different in each component (e.g., from Fig. 3)? What does this entail for different architectures?
3. Are results from Section 5 averaged across positions? Would position representation bias your findings?
4. How is the micro contextualization change (by subtracting a pre-FF attention map) in Section 5.2 different from measuring correlations?

---

> ### Author Response · Authors · 2023-11-20
> **Response to Official Review by Reviewer wwew**
>
> We appreciate your insightful review!
>
> # Limitations of some experiments
> > Weaknesses 1
> > The paper suggests that because of the cancelled out effects in surrounding components there is "potential redundancy in transformer layers". However, there is not enough evidence to justify this claim (e.g., by training a transformer model removing some of the components). Without more experiments and results, it is not convincing what the conclusion of this paper is.
>
> The results in Section 6 revealed that contextualization effects by the Feed-forward network (FF) are canceled by the surrounding components. We agree that this *only suggests the potential redundancy* in the transformer layers. That is why we hedged with the words “potential”, “suggest,” or “imply” in the paper. It would be worthwhile to directly investigate the redundancy of FF, e.g., by observing the model’s behavior change while removing or pruning them. However, we fear that adding such experiments exceeds the focus of a single conference paper. We have refined the manuscript (Section 6) to state the need for further investigation explicitly.
>
> > More importantly, there is no systematic evaluation on the linguistic patterns (e.g., linguistic patterns distribution from different datasets on each layer) apart from some sampled amplified pairs. The results presented in Table 1 and Figure 5 are not evident.
>
> Section 5.2 shows that FF emphasizes specific linguistic compositions, at least across two different models, BERT-base and GPT2 (Figure 5). We agree that the current validation is not fully systematic and this problem can be alleviated by conducting additional experiments on other datasets. We will address this point in the camera-ready version. Nevertheless, it is worth noting that the essential contribution of this study is in part that it provides the new interpretation method and we hope that the acceptance of our study will facilitate more extensive linguistic analyses as you suggested in this community.
>
> # Consistency of results across different sizes and seeds
> > Weaknesses 2
> > The results on different BERT sizes, and different seeds do not seem to be always consistent (e.g., Figure 9, 10 in the appendix).
>
> In Section 5.1, we investigated how much FF, RES2, and LN2 change contextualization in each layer. FF and LN2 tend to change largely in the middle to latter layers in most BERT models with different sizes or seeds. The main claims of this section are: (i) FF block changes contextualization; and (ii) the changes are strong in particular layers rather than even in all layers. At least these points were consistently observed across these models (Figures 9 and 10).
>
> # Reasons for setting a zero vector for the baseline vector in Integrated Gradients
> > Questions 1
> > Why is b set to a zero vector in equation 12?
>
> There are mainly two reasons:
> 1. Using zero vector is one of the most common treatments employed as the neutral baseline for Integrated Gradients [Bastings&Filippova’20; Nayak+’21].
> 2. Since the activation function $g$ in FF satisfies $g(0) = 0$, we can ignore a particular term (output relative to baseline input) when applying Integrated Gradients with the zero baseline vector. This allows Integrated Gradients to fully additively decompose the output into the inputs (Appendix B.2), which is desirable for combining with the norm-based approach.
>
> [Bastings&Filippova’20] The elephant in the interpretability room: Why use attention as explanation when we have saliency methods? (BlackboxNLP 2020) https://aclanthology.org/2020.blackboxnlp-1.14/
> [Nayak+’21] Using Integrated Gradients and Constituency Parse Trees to explain Linguistic Acceptability learnt by BERT (ICON 2021) https://aclanthology.org/2021.icon-main.11/

---

> > ### Author Response · Authors · 2023-11-20
> > **Response to Official Review by Reviewer wwew**
> >
> > # Difference of results between BERT and GPT-2
> > > Questions 2
> > > Why do you think the patterns of changes between BERT and GPT-2 are quick different in each component (e.g., from Fig. 3)? What does this entail for different architectures?
> >
> > At least there are two directions in the differences of BERT–GPT2 results: (i) module level and (ii) layer level.
> > At the module level, contextualization changes by RES are remarkably different between BERT and GPT-2 (Figures 3, 9, and 10). There was almost no contextualization change in BERT, while there was the change in the GPT-2. We suspect that this is due to their different architectures: Post-LN model (BERT) and Pre-LN model (GPT-2). Given that RES in the Pre-LN model bypasses more modules than Post-LN, it is somewhat intuitive that RES in GPT-2 (Pre-LN) overwrites the internal representation more strongly than that in BERT (Post-LN).
> > At the layer level, the contextualization change by FF in BERT is pronounced in the *middle to latter* layers, which is consistent across many variants of BERT (Figures 9 and 10). On the other hand, the change by FF in GPT-2 is pronounced in the *earlier* layers. Understanding this difference may follow the understanding of what each layer does in these models, and the exact interpretation of the model internals is still challenging [Tenney+'19; Elhage+’21]. We hope our study encourages such a deeper level of model understanding. We will add these discussions to the manuscript.
> >
> > [Tenney+‘19]: BERT Rediscovers the Classical NLP Pipeline (ACL 2019) https://aclanthology.org/P19-1452/
> > [Elhage+’21]: A Mathematical Framework for Transformer Circuits (Anthropic) https://transformer-circuits.pub/2021/framework/index.html
> >
> > # Treatment of position in Section 5
> > > Questions 3
> > > Are results from Section 5 averaged across positions? Would position representation bias your findings?
> >
> > We conducted two types of analyses: (i) the macro analysis (Section 5.1), which measures correlations between the attention patterns before and after a particular component; and (ii) the micro analysis (Section 5.2), which analyzes the amplification of attentions between specific types of token pairs. We guess that your concerns are about the latter micro analysis.
> > The FF-amp scores are averaged across the targeted pairs in the corpus; that is, their positions are aggregated following their position distribution in the natural corpus. Further ablation of the position biases (e.g., analyzing with shifting or shuffling input positions) will be an interesting future work.
> >
> > # Measurement of micro contextualization change (Section 5.2)
> > >Questions 4
> > >How is the micro contextualization change (by subtracting a pre-FF attention map) in Section 5.2 different from measuring correlations?
> >
> > In Section 5.2, we aimed to investigate which type of word-to-word interactions are controlled by FFs. For this goal, the single score of correlation between attention maps is not suited since it just summarized the degree of overall similarity between the patterns. Thus, instead of measuring the correlation, we computed the difference of attention maps and observed which cells (word-word pairs) got a large change in Section 5.2.

---

> > > ### Comment · Reviewer_wwew · 2023-11-22
> > >
> > > Thanks for the responses. I have adjusted my evaluation accordingly.

---

### Official Review · Reviewer_Dtpy · 2023-11-10

**Soundness:** 3 good
**Presentation:** 3 good
**Contribution:** 3 good
**Rating:** 8
**Confidence:** 4

**Summary:**

This paper proposes a way of analyzing feedforward blocks (=FFB) [including residual connections and the layer norm layers intervening] extrapolating from the typical attention maps of the mid and late 2010s and the slightly more recent concept of a "refined" attention map that also considers the values and output projection (Equation 8). They call this base formulation ATB. The authors further extend this notion to incorporate FFBs. The FFB's nonlinearity which makes it additively non-decomposable [a-la linearity of expectation] is overcome by using the integration gradients paper [Sundararajan et al, 2017]. The three resultant formulations, which incorporate just the linear, thenceforth residual connections and layer norm are christened ATBFF ATBFFRES and ATBFFRESLN. The Pre-Layer-Norm variants are [like the post-LayerNorm one before] named likewise.

The authors analyze a good variety of encoder only LM checkpoints in addition to a decoder only lM checkpoint [GPT2-117M] , which is also an instance of the pre-LN formulation.

The authors then interestingly proceed to analyze the contextualization changes caused by the FF block , both in terms of extent of change [based on flattening the pairwise values and taking Spearman Correlation of before-after], linguistic contextualization, as well as the dynamics.

**Strengths:**

- Their formalism is extended to both pre and post Layer Norm variants of transformers.
- Tested on a large variety of encoder LM architectures.
- Decoder-only architectures are also covered [though just one, i.e. GPT2-117M].

**Weaknesses:**

- It would have been nice if the authors could have discussed and potentially also experimented with atleast one alternative formulation to IG, or atleast one of its variants [They do mention other formulations in B.1 Appendix but did not see further broaching of this angle beyond this]
- [Doesn't apply after authors comprehensively addressed this on 20th Nov] A marginal weakness but one nonetheless [and this is alluded to in future work], would have been nice to see this for a new-age LLM, of which some variants are available at lower or comparable parametrizations to GPT2-117M (e.g. OPT-125M)

Post-Script: The authors exhaustively addressed the second point! Thanks for that!

**Questions:**

- What is the computational [and memory] complexity of generating these maps? This may sound nitpicky but with increasingly large LLMs which barely fit in the accelerator time and memory bounds whether at training or inference time, this can indeed become a factor and consideration in how widely this gets adapted.
- I know mechanistic explanation is a somewhat orthogonal paradigm of interpreting large transformer architectures, but it would be nice to have some comments on how this can relate or synergize with that paradigm [if at all]
- What is the effect of banded local attention [alternating banded local sparse and global attention] are a common part of the architecural recipe in many GPT3 or later LLMs so this would be a valuable insight to have.

---

> ### Author Response · Authors · 2023-11-20
> **Response to Official Review by Reviewer Dtpy**
>
> We appreciate your invaluable comments.
>
> # Alternatives to Integrated Gradients
> >Weaknesses 1
> >It would have been nice if the authors could have discussed and potentially also experimented with atleast one alternative formulation to IG, or atleast one of its variants [They do mention other formulations in B.1 Appendix but did not see further broaching of this angle beyond this]
>
> Yes, we can use another formulation, such as Shapley values (SV) (Appendix B.1), instead of Integrated Gradients (IG) in our proposed method. At least in the case of using SV, it is required to calculate an expected output change over **all permutations** of input features; that is, SV increases in computational cost more rapidly than IG as input becomes longer. This could especially be a severe problem in modern LLM use, such as providing long prompts/few-shot instances to models and/or considering verbose outputs generated by chain-of-though style reasoning; thus, we used IG as a practical choice.
> In the camera-ready version, we will add this discussion and an experimental comparison of IG and SV for small models in Appendix.
>
> # Application to a new-age LLM
> >Weaknesses 2
> >A marginal weakness but one nonetheless [and this is alluded to in future work], would have been nice to see this for a new-age LLM, of which some variants are available at lower or comparable parametrizations to GPT2-117M (e.g. OPT-125M)
>
> We are starting to experiment on OPT-125M. We may be able to share results during the discussion period. In any case, we plan to add it to the camera-ready version.
>
> # Computational cost of the proposed method
> > Questions 1
> >What is the computational [and memory] complexity of generating these maps? This may sound nitpicky but with increasingly large LLMs which barely fit in the accelerator time and memory bounds whether at training or inference time, this can indeed become a factor and consideration in how widely this gets adapted.
>
> Given an input of length $N$ and FF's intermediate dimension $d$, estimating one attention map requires the computation of Integrated Gradients $Nd$ times. In other words, the computational cost increases with input length and model size. Fortunately, our preliminary analysis shows that the practical computational cost (speed and memory use) seems to be proportional or better than the linear increase with respect to input length and model size ($N$ and $d$) due to some implementation tricks and parallelism (e.g., provided by captum https://captum.ai/).
>
> # Relation to mechanistic explanation
> > Questions 2
> > I know mechanistic explanation is a somewhat orthogonal paradigm of interpreting large transformer architectures, but it would be nice to have some comments on how this can relate or synergize with that paradigm [if at all]
>
> We recognize that mechanistic interpretability/explanation is an attempt to reverse engineer neural networks at the algorithmic level. We believe that our focus, “how the information of a particular token propagates to surrounding tokens in the model (similar to MOV or ADD commands in assembler),” can be an important perspective and provide a contribution there. In particular, our study may facilitate understanding MLPs (feed-forward networks), which has been difficult to interpret mechanistically [Elhage+’22]. We will explicate this point in the camera-ready version (currently, the paper has space limitations, and we will modify the paper after the discussions converge).
>
> [Elhage+’22] Softmax Linear Units (Anthropic) https://transformer-circuits.pub/2022/solu/index.html
>
> # Local sparse and global dense attention in recent LLMs
> > Questions 3
> > What is the effect of banded local attention [alternating banded local sparse and global attention] are a common part of the architecural recipe in many GPT3 or later LLMs so this would be a valuable insight to have.
>
> Thanks for your insightful comment. We will mention this as a future work in the camera-ready version. We hypothesize that the FF affects local and global attention differently; for example, a similar observation as this study (Figure 5) may be observed for local contextualization but, global attention may be affected differently, such as FF selectively expanding specific contextual information that is related to the target token.

---

### Official Review · Reviewer_Ur6S · 2023-11-10

**Soundness:** 4 excellent
**Presentation:** 4 excellent
**Contribution:** 4 excellent
**Rating:** 8
**Confidence:** 3

**Summary:**

The paper proposed methods to analyzed the feed-forward blocks with regards to the input contextualization. It leverages the completeness property of existing norm-based analysis and the integrated gradient method.

The motivation to analyze the feed-forward blocks include the following:
1. The feed-forward blocks account for 2/3 of the layer parameters.
2. There is a growing interest in feed-forward blocks (new approaches focusing on the feed-forward blocks)
3. Previous work reported that feed-forward blocks perform some linguistic operations

Their experiments using masked-LM and casual-LM have shown that feed-forward blocks modify the input contextualization by amplifying specific types of linguistic compositions. Feed-forward blocks and layer normalization largely control contextualization.

They also found that feed-forward block and other blocks cancel out each other's contextualization effects, which might indicates redundancy in the Transformer computations. (Feed-forward blocks' effects are weekend by surrounding residual and normalization layers.)

**Strengths:**

The paper is the first to analyze the whole feed-forward blocks, including the non-linear activation function. The non-linear activation function has been previously excluded from the norm-based analyses because it cannot be decomposed additively by the distributive law.

Combining the norm-based and the integrated gradients, the paper is able to quantify and visualize the effects of the whole feed-forward block.

The paper also provides detailed analysis that points to interesting properties of the feed-forward blocks. The discovery of redundancy might leed to new improvement of the architecture.

**Weaknesses:**

The author pointed out that the future work might be working with the latest large language model.

**Questions:**

I think it might help the reader with a description or definition of contextualization.

---

> ### Author Response · Authors · 2023-11-20
> **Response to Official Review by Reviewer Ur6S**
>
> We appreciate your insightful feedback and positive reviews.
>
> # Application to the latest large language models
> >Weaknesses
> >The author pointed out that the future work might be working with the latest large language model.
>
> We agree that this direction is an attractive future work, given that the LLMs’ internal mechanisms have been less understood. Our method can be extended to LLMs, and as a first step, we are starting experiments with OPT-125M (we may be able to share results during the discussion period). We will append these results in the camera-ready version at the latest.
>
> # Description or definition of contextualization
> >Questions
> >I think it might help the reader with a description or definition of contextualization.
>
> We have clarified what is referred to by “contextualization” in the second paragraph of the Introduction (Section 1) and the second paragraph of the Background (Section 2).

---

### Author Response · Authors · 2023-11-21
**General response to reviewers**

# Results for OPT-125M are consistent with the other models
> Weaknesses by Reviewer Ur6S
> The author pointed out that the future work might be working with the latest large language model.

> Weaknesses 2 by Reviewer Dtpy
> A marginal weakness but one nonetheless [and this is alluded to in future work], would have been nice to see this for a new-age LLM, of which some variants are available at lower or comparable parametrizations to GPT2-117M (e.g. OPT-125M)

Following the constructive feedback above, we have conducted some of our experiments with OPT-125M and added the results to the Appendix of the manuscript. We can observe the consistent trends between OPT and the other models, enhancing the generalizability of our experiments:
- Macro contextualization changes by LN, FF, and RES (experiments in Section 5.1) in OPT-125M are shown in Figure 12 (Appendix C.1). The results for OPT-125M show consistent trends with the other models: FFs and LNs in particular layers incur strong contextualization changes.
- Contextualization changes through FF and subsequent components (experiments in Section 6) in OPT-125M are shown in Figure 17 (Appendix D.1). The results for OPT-125M show consistent trends with the other models: through the FF and subsequent RES and LN, the score once becomes large but finally converges to be small (cancellation of contextualization changes).

We have also slightly updated the descriptions in Sections 4, 5, and 7. We will conduct the other experiments and add the results in the camera-ready version.

Please see the reply to each review for our responses to the other comments/feedback/questions.

---

> ### Comment · Reviewer_Dtpy · 2023-11-21
> **That's great!**
>
> Thanks for the insightful update and running this impromptu at short notice ... this makes the range of models covered much more solid

---

### Meta-Review · Program_Chairs · 2023-12-06

**Metareview:**

This is a clearly written, analysis paper that investigates the feed-forward blocks of transformers for input contextualization effects. The findings from the analysis, especially regarding the redundancy can inspire new future contributions. Reviewers have made a set of suggestions that were possible to address during the rebuttal, such as including experimentation with larger models and extending the related work, and the authors responded positively.

**Justification For Why Not Higher Score:**

Limitations still exist.

**Justification For Why Not Lower Score:**

This paper could inspire some good discussions and future investigations.

---

### Decision · Program_Chairs · 2024-01-16

Accept (spotlight)